# Time cell sequences during delay intervals are not dependent on brain state and do not support hippocampus-dependent working memory

Li Yuan[1], Jose F. Figueroa [1], Ameen Khan[1], Gautam Narayan [1], Jill K. Leutgeb [1,2,3] ✉ & Stefan Leutgeb [1,2,3] ✉

Working memory (WM) is essential for performing cognitive tasks, and sequentially active hippocampal cells over many seconds ('time cells') have been observed during WM retention. Time cells predominantly occur when neural activity oscillates at theta frequency. To examine whether time cells during WM maintenance depend on ongoing theta oscillations, we controlled the persistence of theta during 10 s and 30 s delay intervals by either having rats run or rest, which resulted in conditions with and without persistent theta oscillations. In either condition, reliable time cells were limited to only the first few seconds of the delay interval while a second population of constitutively active cells emerged during the remainder of the delay period, neither of which were memory-related. Our results show that hippocampal sequential activity patterns are short-lasting and uninformative for WM, and that WM retention over more than ~5 s needs to include mechanisms other than hippocampal time cells.

Working memory (WM) over periods of seconds is essential for the performance of cognitive tasks. Various types of neuronal activity over delay intervals in WM tasks are thought to serve as potential neural mechanisms for information retention. For example, sustained spiking during delay intervals was first reported in dorsolateral prefrontal cortex (PFC) of primates[1,2], and subsequent studies also identified delay-active cells in various other brain regions, such as the frontal eye field[3,4], the anterior cingulate cortex[5], the orbitofrontal cortex[6], and inferotemporal cortex[7]. In addition, sequential rather than sustained neural firing has been observed during delay intervals in prefrontal areas and in other memory areas, such as hippocampal and entorhinal cortex[8–12]. Sequential hippocampal activity patterns during delay intervals have been shown to correlate with successful behavior outcomes in WM tasks that depend on hippocampus and entorhinal cortex[8,13–16]. In particular, studies that reduced hippocampal theta oscillations—the predominant local field potential (LFP) during

running[17]—demonstrated that the extent of remaining sequential firing was closely related to memory performance[18,19]. However, similar subsequent studies also assessing population activity in WM tasks could not detect a relationship between hippocampal delay activity and spatial WM performance[16,20,21]. The studies that could not detect memory-related firing patterns were all performed without a requirement to run during the delay interval, which may therefore correspond to the critical condition for making hippocampal spiking activity during delay intervals informative for WM.

While rodents are running, ongoing theta oscillations are accompanied by temporal ordering of hippocampal neuronal activity within a theta cycle ('theta sequences'), which is a time-compressed version of sequential activity patterns on a behavioral time scale[17,22,23]. Even in a setup where rats were immobilized in a WM task and thus not actively running, the sequential activity during the delay interval ('time cells') had a strong temporal relationship with co-existing theta

[1]Neurobiology Department, School of Biological Sciences, University of California San Diego, La Jolla, CA, USA. [2]Institute for Advanced Study, Berlin, Germany. [3]Kavli Institute for Brain and Mind, University of California San Diego, La Jolla, CA, USA. ✉e-mail: jleutgeb@ucsd.edu; sleutgeb@ucsd.edu

oscillations[24]. Despite the possible relation between theta oscillations and sequential firing over time scales of many seconds, reports on the effects of reducing theta oscillations on time cells are inconsistent. Hippocampal time cells are diminished when theta amplitude is substantially reduced by pharmacological silencing of the medial septal area (MSA)[18], but retained when theta amplitude is reduced with optogenetic silencing of MSA[19]. Similarly, in spatial alternation tasks with no requirement for animals to run during the delay interval and thus with reduced theta oscillations, hippocampal time cell sequences last only for a few seconds[16]. Evidence on the persistence of time cells at low levels of theta oscillations is therefore inconsistent, and these studies have not established whether increasing or prolonging theta oscillations is necessary for extending the time period over which sequential activity is sustained during WM.

To determine whether sustained running and the accompanying continuous theta oscillations prolong time cell sequences or whether hippocampal network mechanisms other than sequential activity patterns emerge without running and/or in longer delay intervals, we examined the emergence, persistence and information content of time cells during 10-s and 30-s delay intervals. In addition, the continuity of theta oscillations during the delay period was manipulated by leveraging the established relationship between movement and hippocampal theta activity. By recording from hippocampal CA1 ensembles in these varied WM task conditions, we were able to identify potential mechanisms that support WM over delay intervals of different lengths and with different hippocampal oscillation patterns.

## Results

### Behavior during the delay determined the persistence of theta oscillations

To test to what extent the neural code in delay intervals during a WM task is determined by ongoing theta oscillations, we designed a delayed alternation task with trials when rats were either forced to run continuously on a treadmill ('treadmill on') or were allowed to rest during the delay interval ('treadmill off'; Fig. 1a). When running continuously, the head remained in a consistent position, and by design, running speed throughout the entire delay interval corresponded to the treadmill speed (Fig. 1b, c, Fig. S1). The delay length was either 10 s or 30 s, such that there were four combinations of delay length and treadmill condition. The four delay/treadmill combinations were used over eight blocks of 10 trials, with each combination repeated two times in the same recording session (Fig. 1a). Irrespective of the substantial differences in theta oscillation patterns and delay lengths, rats performed the alternation task with approximately equally high accuracy in each condition (percent correct, on/10 s: $87.2\% \pm 2.70$, off/10 s: $93.1\% \pm 1.76$, on/30 s: $86.9\% \pm 2.36$, off/30 s: $85.1\% \pm 2.42$, mean ± SEM, $n = 18$ sessions from 5 rats, F(3,68) = 2.21, $p = 0.09$, ANOVA; Fig. 1d).

Because of the link between movement and the emergence of hippocampal theta, we expected that running on the treadmill controlled the power and persistence of theta oscillations during the delay interval. High-amplitude theta oscillations were observed in all 5 rats during forced running throughout the 10-s and 30-s delay intervals (Fig. 1e–h, Fig. S2 and S3), such that theta power was higher and theta bouts were longer when rats were continuously running with the treadmill on compared to resting with the treadmill off (z-scored theta power, on/10 s: $0.53 \pm 0.06$, off/10 s: $-0.05 \pm 0.07$; on/30 s: $0.55 \pm 0.07$, off/30 s: $-0.01 \pm 0.06$, mean ± SEM, $n = 18$ sessions in 5 rats; treadmill on vs. off, 10 s delay: $p = 1.1 \times 10^{-7}$; 30 s delay: $p = 7.7 \times 10^{-12}$, paired $t$-tests; all theta bouts, on/10 s: $5.15 \pm 0.47$ s, off/10 s: $2.79 \pm 0.19$ s, on/30 s: $8.54 \pm 1.30$ s, off/30-s: $3.59 \pm 0.51$ s, mean ± SEM, $n = 18$ sessions; F(3,68) = 12.51, $p = 1.3 \times 10^{-6}$, ANOVA; initial theta bouts, on/10 s: $5.28 \pm 0.45$ s, off/10 s: $2.86 \pm 0.28$ s, on/30-s: $7.94 \pm 1.18$ s, off/30 s: $3.37 \pm 0.51$ s, mean ± SEM, $n = 18$ sessions; F(3,68) = 11.66, $p = 3.0 \times 10^{-6}$, ANOVA). In addition, delay intervals included longer periods with

sustained theta oscillations when the treadmill was on rather than off during the delay (% time with theta: on/10 s: $81.5\% \pm 3.24$, off/10 s: $66.7\% \pm 2.89$, on/30 s: $83.9\% \pm 2.94$, off/30 s: $67.9\% \pm 2.99$, mean ± SEM, $n = 18$ sessions; F(3,68) = 9.32, $p = 3.0 \times 10^{-5}$, ANOVA; percent trials with >80% of delay duration in theta: on/10 s: $65.0\% \pm 7.13$, off/10 s: $29.3\% \pm 5.36$, on/30 s: $71.3\% \pm 8.06$, off/30 s: $24.2\% \pm 6.95$, mean ± SEM, $n = 18$ sessions; F(3,68) = 12.81, $p = 1.0 \times 10^{-6}$, ANOVA; Fig. 1i, j).

### The delay area was overrepresented by CA1 cells irrespective of treadmill condition and delay length

After confirming that the persistence of theta bouts substantially differed between delay intervals with and without sustained running, we asked whether hippocampal neuronal activity patterns during the delay interval depended on ongoing theta oscillations. We recorded 377 CA1 putative pyramidal neurons while rats performed the delayed alternation task. All combinations of treadmill and delay conditions were included in each recording session ($n = 18$ sessions from 5 rats), and only cells with stable recordings from the beginning to the end of the entire recording session were used for further analysis. In each treadmill/delay condition, the proportion of putative pyramidal cells that were active (average rate > 0.5 Hz) in maze areas other than the delay zone was ~1.5 fold higher than the proportion active in the delay zone (% cells active in remainder of the maze, on/10 s: $45.9\% \pm 5.0$; off/10 s: $42.1\% \pm 4.6$; on/30 s: $42.6\% \pm 5.2$; off/30 s: $43.5\% \pm 5.1$; % cells active in delay, on/10 s: $32.3\% \pm 3.2$; off/10 s: $34.6\% \pm 4.2$; on/30 s: $31.2\% \pm 4.1$; off/30 s: $34.6\% \pm 4.9$; mean ± SEM, $n = 18$ sessions from 5 rats; Fig. 2a). In comparison, the size of the remainder of the maze was ~4-fold larger than the delay zone (Fig. 2b). Therefore, the proportion of cells active in the delay area was higher than predicted from the relative size of the delay zone (delay zone, ratio of % active cells by % size of zone, on/10-s: $1.6 \pm 0.16$; off/10-s: $1.7 \pm 0.21$; on/30-s: $1.6 \pm 0.20$; off/30-s: $1.7 \pm 0.24$, mean ± SEM; remainder of the maze: on/10-s: $0.58 \pm 0.07$; off/10 s: $0.53 \pm 0.07$; on/30 s: $0.53 \pm 0.08$; off/30-s: $0.55 \pm 0.07$, mean ± SEM; F(1,143) = 112.48, $p = 1.6 \times 10^{-19}$, $n = 18$ sessions from 5 rats, ANOVA: two-factor with replication), similar to the pattern that has been described in the running wheel[8]. The overrepresentation of the delay area could be explained by the amount of time in the delay zone rather than the spatial size of the delay zone. However, irrespective of the proportion of time that was spent in the delay zone (10 s delay: $26.6\% \pm 0.92$; 30 s delay: $50.0\% \pm 1.0$; mean ± SEM; $n = 18$ sessions; Fig. 2b), the proportion of delay-active cells remained approximately the same (F(3,68) = 0.18, $p = 0.91$, ANOVA; Fig. 2a). Accordingly, the fraction of delay-active cells was independent of delay length, and given the similar proportions between treadmill-on and off conditions, also independent of the prevalence of theta oscillations. We next asked whether delay-active cells also showed spatially selective firing outside of the delay zone and found that many of the delay-active cells had place fields throughout the remainder of the maze (Fig. 2c, Fig. S4). Within the delay, the peak times of delay-active cells were distributed over the entire delay interval (Fig. 2d). The firing patterns of delay-active cells were investigated in more detail.

### Delay-active cells differed between treadmill on and off rather than across delay durations

With the proportion of delay-active cells matching across conditions, we next asked whether the same or different cell populations were active in each of the conditions (see Fig. 3a for example cells and Fig. 3b for all cells). There was a large overlap in the identity of delay-active cells across the 10 s and 30 s delay conditions (91.3% of cells for treadmill on, 86.5% for treadmill off; Fig. 3c), but a smaller overlap across treadmill-on and off conditions (66.9% of cells with 10 s delay and 62.2% with 30-s delay). Similar to the proportions, the firing rates of active cells were more similar across delay conditions than across treadmill conditions (difference over sum, $0.22 \pm 0.01$ and $0.5 \pm 0.02$, $p = 1.0 \times 10^{-22}$, Kolmogorov-Smirnov test; mean firing rates, 10 s vs. 30 s

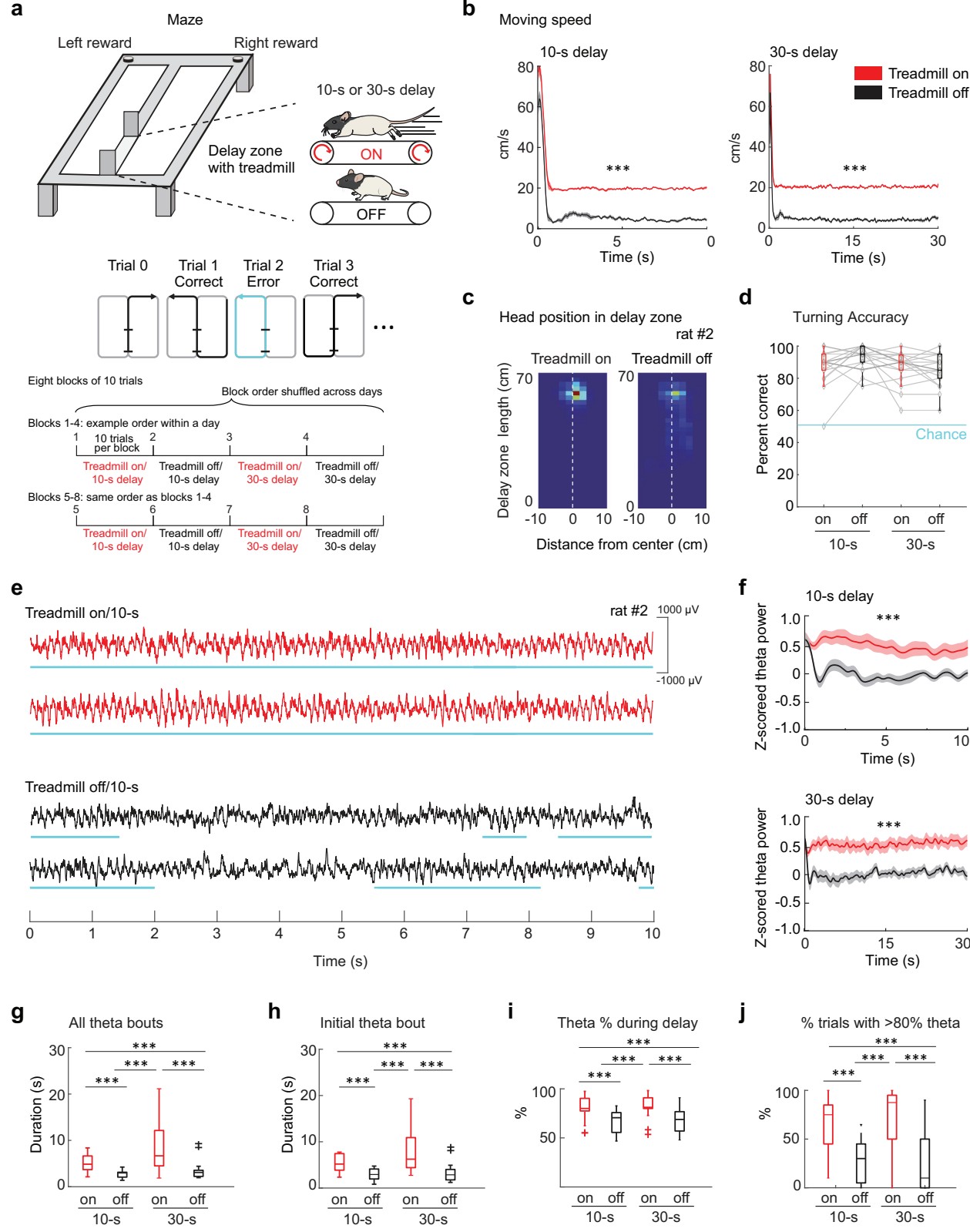

delay, treadmill on: Spearman's $r = 0.91$, treadmill off: $r = 0.82$; treadmill on vs. off, 10 s delay: $r = 0.21$; 30 s delay: $r = 0.13$; correlation between delay conditions vs. between treadmill conditions, $p < 0.001$, Fisher z-transformation; Fig. 3d, e). Taken together, cells' firing rates and patterns in the delay area corresponded well between 10 s and 30 s delays, but to a lesser extent across conditions with different treadmill conditions.

## Time cell firing patterns showed a major transition after ~ 5 s

To better understand why the increase in delay length was not coupled with a recruitment of additional hippocampal cells during longer delay intervals, we performed additional comparisons of the neuronal activity patterns between the 10-s and the 30-s delay trials. We focused on cells that were active for a defined time period during the delay interval, which have previously been reported in the hippocampal CA1

**Fig. 1 | Resting or running on a treadmill during the delay determined the amplitude and persistence of theta oscillations. a** Schematic of the spatial WM task. The rat's behavior during delay intervals was controlled by a treadmill, which was either on or off for a 10 s or 30 s delay period. On each day, the order of the first four treadmill/delay blocks (on/10 s, off/10 s, on/30 s, and off/30 s; 10 trials per block) was shuffled, and the four blocks were repeated for a second time in the same order. **b** Movement speed during delay intervals (on/10 s: 22.15 ± 0.26 cm/s, off/10 s: 7.49 ± 0.43 cm/s, on/30 s: 21.32 ± 0.23 cm/s, off/30 s: 5.46 ± 0.42 cm/s; mean ± SEM, $n = 18$ sessions in 5 rats; treadmill on vs. off, 10 s delay: $p = 4.0 \times 10^{-20}$; treadmill on vs. off, 30 s delay: $p = 1.0 \times 10^{-19}$, paired $t$-tests). **c** Head position within the delay zone during treadmill-on (left) and treadmill-off (right) trials. Heatmaps report the occupation density (blue, zero; red, maximum) of the head position of an example rat across treadmill-on trials ($n = 80$ on/30 s trials from 4 sessions) and treadmill-off trials ($n = 80$ off/30 s trials from 4 sessions). See Figure S1 for data from additional rats. **d** Spatial WM task performance in different treadmill/delay conditions. No differences were found between conditions (on/10 s: 87.2% ± 2.70, off/10 s: 93.1% ± 1.76, on/30 s: 86.9% ± 2.36, off/30 s: 85.1% ± 2.42; mean ± SEM, $n = 18$ sessions from 5 rats; F(3,68) = 2.21, $p = 0.09$, ANOVA). **e** Raw LFP traces (10 s each) from the same recording site in 10 s delay trials with the treadmill either on (red) or off (black). Blue lines, periods of continuous theta oscillations. **f** Theta amplitude was higher in treadmill-on (red) than in treadmill-off (black) delay periods (top: z-scored theta power, 10 s delay, on: 0.53 ± 0.06, off: −0.05 ± 0.07; bottom: 30 s delay, on: 0.55 ± 0.07, off: −0.01 ± 0.06, mean ± SEM, $n = 18$ sessions in 5 rats; treadmill on vs. off, 10 s delay: $p = 1.1 \times 10^{-7}$; 30 s delay: $p = 7.7 \times 10^{-12}$, paired $t$-tests). **g** Average theta bout duration was longer in treadmill-on compared to off conditions (on/10 s: 5.15 ± 0.47 s, off/10 s: 2.79 ± 0.19 s, on/30 s: 8.54 ± 1.30 s, off/30 s: 3.59 ± 0.51 s, mean ± SEM, $n = 18$ sessions; F(3,68) = 12.51, $p = 1.3 \times 10^{-6}$, ANOVA). **h** The duration of the initial theta bout during the delay was longer in treadmill-on compared to off conditions (on/10 s: 5.28 ± 0.45 s, off/10 s: 2.86 ± 0.28 s, on/30 s: 7.94 ± 1.18 s, off/30 s: 3.37 ± 0.51 s, mean ± SEM, $n = 18$ sessions; F(3,68) = 11.66, $p = 3.0 \times 10^{-6}$, ANOVA). **i.** Percentage of the total delay period with high theta power was higher in treadmill-on compared to off conditions (on/10 s: 81.5% ± 3.24, off/10 s: 66.7% ± 2.89, on/30 s: 83.9% ± 2.94, off/30 s: 67.9% ± 2.99, mean ± SEM, $n = 18$ sessions; F(3,68) = 9.32, $p = 3.0 \times 10^{-5}$, ANOVA). **j** Percentage of trials with sustained theta (theta periods > 80% of total delay duration) was higher with the treadmill on than off (on/10 s: 65.0% ± 7.13, off/10 s: 29.3% ± 5.36, on/30 s: 71.3% ± 8.06, off/30 s: 24.2% ± 6.95, mean ± SEM, $n = 18$ sessions; F(3,68) = 12.81, $p = 1.0 \times 10^{-6}$, ANOVA). All statistical tests are two-sided without post-hoc adjustments for multiple comparisons. Box plots: central line, edges, whiskers and plus signs indicate median, the 25th/75th percentile, maximum/minimum and outliers. *** $p < 0.001$. Source data are provided as a Source Data file.

area in tasks with and without running during a delay interval ('sequence cells' or 'time cells')[8,9,16,25]. We identified time cells by examining whether a cell's neuronal activity was consistently timed across trials. For an unbiased classification, we calculated each cell's firing rates over 150-ms time bins throughout the delay interval, obtained the correlation across each pair of trials, and compared the correlation values of the real data to distributions calculated from shuffling the time bins (Fig. 4a). As expected, we found cells with consistently timed firing profiles throughout the 10-s and 30-s delay intervals (Fig. 4b), and the peak times of these cells spanned the entire delay interval (Fig. 4c, d, see Fig. S5 for individual rats). As previously reported[9,26], the length of the time period when cells were active ('time field size') increased for cells that were active late in the delay interval. However, we noted that many of the broader time fields during the second half of the delay extended to the end of the delay, which raised the question whether they exhibited qualitatively different activity patterns. To examine this possibility without a priori classifying the cells, we compared the firing profiles of time cells within and between the 10-s and 30-s delay intervals. Cells with narrow time fields early in the 10 s delay interval showed approximately corresponding activity patterns early in the 30 s delay interval. As a consequence, the firing rates during the first 5 s of the 10 s delay and the 30 s delay were highly correlated (treadmill on: $r^2 = 0.61$, Pearson's linear correlation; treadmill off: $r^2 = 0.68$, Pearson's linear correlations; Fig. 4c, d). In contrast, firing rates of cells substantially changed after the first few seconds (correlation between the 0–5 s and 5–10 s intervals, on/10 s delay: $r^2 = -0.49$; on/30 s delay: $r^2 = -0.30$; off/10 s delay: $r^2 = -0.26$; off/30 s delay: $r^2 = -0.46$, Pearson's linear correlation). In the 30 s delay, cells that were active in the 5−10 s interval then often continued to fire for the remainder of the delay (treadmill on: 5–10 s vs. 10–15 s: $r^2 = 0.51$; vs. 15−20 s: $r^2 = 0.48$; vs. 20-25 s: $r^2 = 0.21$; vs. 25-30 s: $r^2 = 0.24$, Pearson's linear correlations; treadmill off: 5–10 s vs. 10–15 s: $r^2 = 0.43$; vs. 15−20 s: $r^2 = 0.32$; vs. 20−25 s: $r^2 = 0.20$; vs. 25−30 s: $r^2 = 0.07$, Pearson's linear correlations; Fig. 4c, d, and Table S1). These results are not consistent with the hypothesis of sequentially active cells throughout the entire delay interval and rather suggest that there are two separate types of activity patterns. Sequential firing of time-limited cells spans the first few seconds, while persistently active cells turn on after a few seconds and remain active for the remainder of the delay interval. The emergence of persistent firing fits with the observation that the fraction of delay-active cells did not further increase in the 30 s delay compared to the 10 s delay (see Fig. 2a)−most cells that were active in the 5−10 s interval remained active for the remainder of the delay (see

Fig. 4c, d). Because these results are consistent with two distinct temporal firing profiles, we further divided time cells into time-limited cells (firing rate < 20% of peak rate in any time bin between the peak and the end of the delay) and into persistently active cells (firing rate ≥ 20% of peak rate in all time bins between the peak and the end of the delay).

## Time-limited cells
We identified a similar fraction of time-limited cells in all delay/treadmill conditions (on/10 s: 11.2% ± 2.19; off/10 s: 9.1% ± 2.4; on/30 s: 10.7% ± 1.89; off/30 s: 12.2% ± 1.92, mean ± SEM, $n = 18$ sessions; F(3,68) = 0.41, $p = 0.75$, ANOVA; Fig. S6a, b, Table S2 and S3), and most time-limited cells had finely tuned time fields with peak firing during the initial 5 s of the delay interval (on/10-s: 98.5% ± 1.6; off/10 s: 100.0% ± 0.0; on/30 s: 84.9% ± 7.92; off/30 s: 82.8% ± 8.54, mean ± SEM, $n = 18$ sessions; Fig. 5a−c, Fig. S6c, d). Our experimental conditions differed from earlier reports that showed time-selective sequential firing for ~15 seconds of wheel/treadmill running[8,18,25] in that we tested each rat not only with running during the delay, but in interleaved blocks of trials also in a condition without running during the delay. To therefore examine whether our results might have emerged from training with interleaved treadmill-on and off conditions, we added two additional experimental groups (Fig. S7). Both additional groups were trained in the same apparatus as the group with the combined training, but one was trained with only the treadmill on during all trials ($n = 16$ sessions in 4 rats) while the other was trained with only the treadmill off during all trials ($n = 25$ sessions in 4 rats). When comparing the fraction of time-limited cells across training groups, there was a lower proportion in the group with only treadmill-on training compared to the group with combined training, and there were no differences between the groups with only treadmill-off training and combined training (treadmill-on rats, 10 s: 6.2% ± 2.21, 30 s: 6.7% ± 2.16; treadmill-off rats, 10 s: 10.8% ± 2.07, 30 s: 10.2% ± 2.24; treadmill-on and off rats, on/10 s: 11.2% ± 2.19, on/30 s: 10.7% ± 1.89, off/10 s: 9.1% ± 2.40, off/30 s: 12.2% ± 1.92, mean ± SEM; Fig. S6b and see Table S2 for statistics). The fraction of time-limited cells in rats trained with both treadmill conditions was therefore higher or comparable to the fraction in rats trained with only one of the treadmill conditions. Irrespective of the training protocol, peak firing times were always concentrated early in the delay interval (i.e., >82% of all time-limited cells in the first 5 s; Fig. 5c). Furthermore, peak times were consistent between the 10 s and 30 s delay conditions across all training groups (treadmill-on rats: $n = 15$ cells, Spearman's $r = 0.91$, $p = 2.1 \times 10^{-6}$;

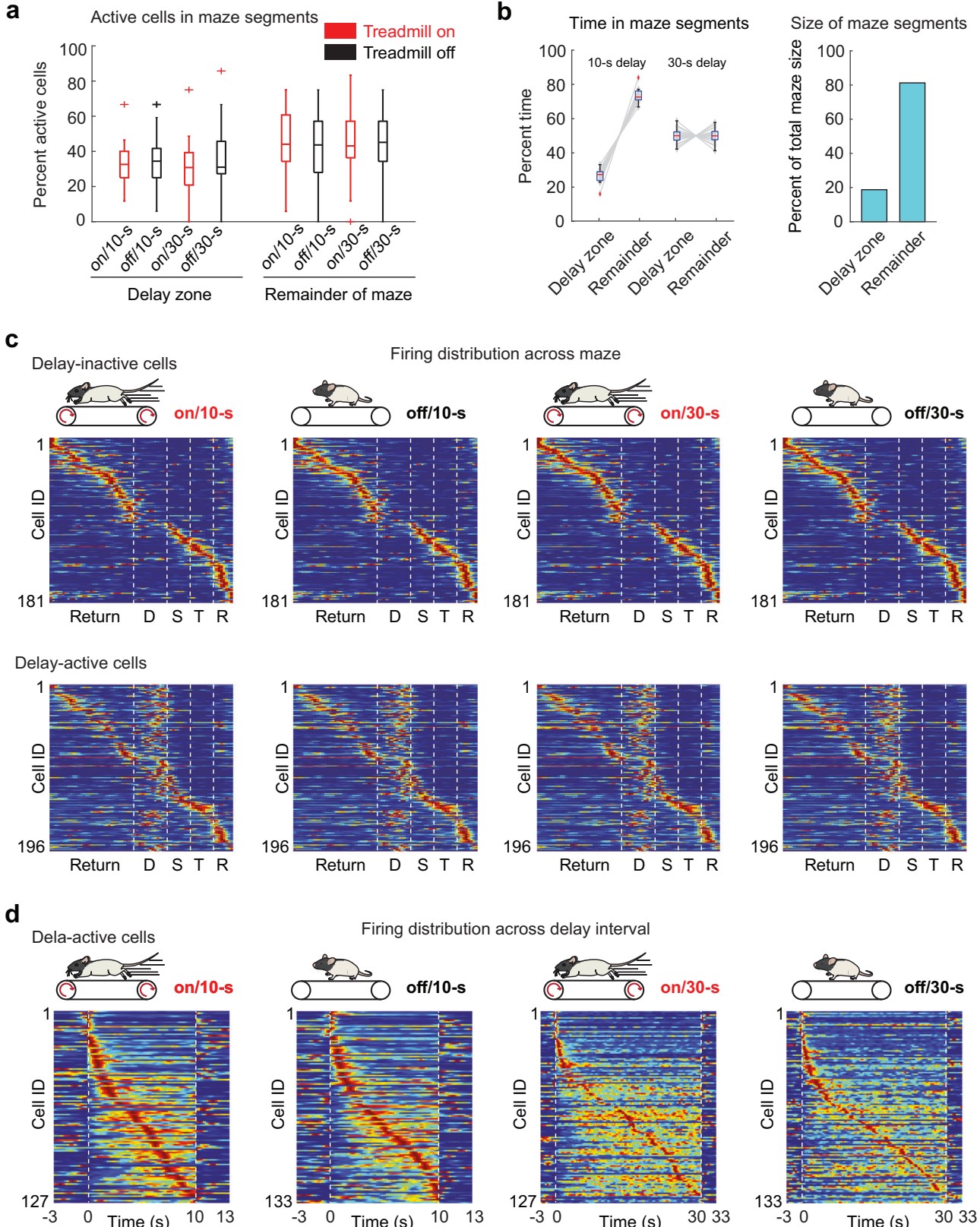

**a** Active cells in maze segments

**b** Time in maze segments · Size of maze segments

**c** Firing distribution across maze
Delay-inactive cells
on/10-s · off/10-s · on/30-s · off/30-s
Delay-active cells

**d** Firing distribution across delay interval
Dela-active cells
on/10-s · off/10-s · on/30-s · off/30-s

treadmill-off rats: $n = 28$, Spearman's $r = 0.82$, $p = 1.0 \times 10^{-7}$; treadmill-on and off rats, treadmill-on condition: $n = 46$ cells, Spearman's $r = 0.85$, $p = 1.3 \times 10^{-14}$, treadmill-off condition: $n = 40$ cells, Spearman's $r = 0.79$, $p = 5.9 \times 10^{-10}$; Fig. S6d and see Table S2 for statistics). The prevalence of time-limited cells with time fields during only the first few seconds is therefore not a phenomenon that is limited to the

combined training protocol, but was also observed when exclusively training with or without running during the delay period.

We next examined whether the low fraction of time-limited cells and the early peak times in our study may have been the consequence of a more rigorous criterion than in earlier studies, and for comparison, applied the criterion from Ref. 8 to our data. With their criterion,

**Fig. 2 | Resting or running on a treadmill during the delay did not change the proportion of delay-active cells or the distribution of their activity.**
**a** Proportion of putative pyramidal cells that were active (average rate > 0.5 Hz) in the delay zone (by treadmill/delay condition, on/10-s: 32.3% ± 3.23; off/10 s: 34.6% ± 4.23; on/30 s: 31.2% ± 4.08; off/30 s: 34.6% ± 4.88; mean ± SEM, $n = 18$ sessions; $p = 0.91$, F(3,68) = 0.18, ANOVA) or in maze areas other than the delay zone (on/10 s: 45.9% ± 4.98; off/10 s: 42.1% ± 4.56; on/30 s: 42.6% ± 5.16; off/30-s: 43.5% ± 5.07, mean ± SEM, $n = 18$ sessions; $p = 0.94$, F(3,68) = 0.13, ANOVA).
**b** Left, percent time spent in the delay zone compared to the remainder of the maze (10 s delay: 26.6% ± 0.92 and 73.4% ± 0.92; 30 s delay: 50.0% ± 1.04 and 50.0% ± 1.04, mean ± SEM; $n = 18$ sessions). Right, percent of the total figure-eight maze area comprised by the delay zone or other maze locations. **c** Top, spatial distribution of delay-inactive cells' activity patterns ($n = 181$ principal neurons across all treadmill/

delay conditions) on the maze in each delay condition. Bottom, spatial distribution of delay-active cells' activity patterns ($n = 196$ principal neurons across all treadmill/delay conditions) on the maze in each delay condition. Cells are sorted by the peak firing location across the average over the four conditions. Blue, 0 Hz; red, each cell's maximum firing rate within a condition; D, delay zone; S, stem; T, T zone; R, reward zone. **d** Delay-active cells in each delay condition, sorted by their peak firing time within each treadmill/delay condition (number of active cells in on/10 s, off/10 s, on/30 s, and off/30 s: $n = 127, 133, 127$, and 133 of 377 principal neurons; $p = 0.94$, chi-square test). Blue, 0 Hz; red, each cell's maximum firing rate. All statistical tests are two-sided without post-hoc adjustments for multiple comparisons. Box plots: central line, edges, whiskers and plus signs indicate median, the 25th/75th percentile, maximum/minimum and outliers. Source data are provided as a Source Data file.

we detected an even lower fraction of time cells than with our shuffling method (4.9–7.7% across treadmill/delay conditions, $n = 5$ rats, Fig. S8a and see Table S3 for statistics), and again, the cells that were detected included a high fraction of cells with peak times in only the first few seconds (84.9–100.0%). Finally, we reasoned that the low fraction of time cells compared to other studies could be a consequence of extensive training experience in our cohort. Therefore, we added an additional experimental group ($n = 19$ sessions in 6 rats). In these rats, we recorded CA1 cells with chronically implanted Neuropixels probes, which allowed us to perform recordings with a more limited number of training days. These recordings revealed a much higher proportion of time cells in rats with less training (38.1-54.6%, $n = 6$ rats, compared to 15.0-20.0%, $n = 5$ rats; Fig. S8b, c), and again, included a high fraction of time-limited cells of which most had their peak times in the first few seconds (90.9-100.0%).

## Time-dependent firing patterns were unrelated to theta power
The similarity in the proportion of time-limited cells in the treadmill-on compared to the treadmill-off conditions (Fig. S6b) already suggested that ongoing theta oscillations during the delay interval did not bolster the emergence of sequentially active cells. To further investigate differences in time cell properties with different theta persistence, we tested effects on time cell stability and on the distribution of peak times. Despite the substantial differences in the persistence of theta oscillations between treadmill conditions, we did not find any differences in time field stability or in the distribution of peak times (Fig. 5c). As expected, time field stability decreased for time fields with later peak times, but the effect was observed to the same extent in treadmill-on and off conditions (Fig. 5d). In addition, we checked whether it might require long periods of uninterrupted theta in treadmill-on conditions to elicit more prolonged time cell sequences and examined only trials with theta bouts over at least 80% of the delay period. Again, the same pattern as for the complete dataset emerged, with peak times of time-limited cells predominantly in the first few seconds (Fig. S9). Overall, our results therefore suggest that theta oscillations neither extended the period when time-limited cells were prevalent nor stabilized their trial-to-trial temporal consistency.

## Persistently active cells
Similar to what we observed for time-limited cells, the fraction of persistently active cells did also not differ between treadmill-on and off conditions (on/10 s: 8.9% ± 1.78, off/10 s: 9.8% ± 2.50, $p = 0.74$, paired $t$-tests, $n = 18$ sessions; on/30 s: 5.1% ± 1.28, off/30 s: 3.7% ± 0.93, $p = 0.39$, paired $t$-tests, $n = 18$ sessions; Fig. 6a–d). To examine whether cells that were identified in the 10 s condition as persistently active (i.e., firing until the end of the 10 s delay) were also active until the end of the 30 s delay, we quantified the firing rates of persistently active cells across delay conditions. These data show that the cells' firing rates at the end of the 10-s delay and at the end of the 30 s delay were strikingly similar ($r = 0.829$, $n = 37$, $p = 5.5 \times 10^{-8}$, Spearman correlation; Fig. 6e). Persistently active cells typically turned on after ~5 s (median onset time:

4.7 s, with onset time defined as time when first exceeding 20% of peak rate) and then continued to be active for the remainder of the delay interval. In fact, it can be assumed that the original identification of these cells as time cells almost entirely resulted from their relative silence early in the delay period, as these cells generally did not have any further temporal selectivity once they started firing (see Fig. 6a–c). Apart from the brief period of initial silence, the persistently active cells are thus similar to the large fraction of delay-active cells (Figs. 2d and 3b) for which no temporal selectivity could be detected. Without temporal selectivity and sequential activity on the behavioral time scale, it can be predicted that persistently active cells are less likely to show phase-precession. As expected, the fraction of persistently active cells with phase precession (~20%) was substantially lower than the fraction of time-limited cells with phase precession (~50%; Fig. S10).

## Neither single cells nor cell assemblies showed trajectory-dependent coding during the delay interval
Cells in hippocampus have been reported to code for past or future trajectories while rats were running on the stem or in a wheel next to the stem[8,27]. However, trajectory-dependent coding in delay zones has been reported to be less prominent or not detectable when rats were not forced to run[13,16,20]. It is therefore possible that trajectory-dependent coding selectively emerges only with running that results in ongoing theta oscillations. Given that we included delays with and without continuous running in the same session in our task, we were able to directly compare trajectory-coding across the conditions with different movement patterns. To measure trajectory coding during the delay for single CA1 cells, we compared the firing rate in the delay between trials with either upcoming right turns or left turns. By comparing the data to a distribution that was obtained with shuffled turn directions, we identified whether the coding for turn direction was significant (i.e., > 95th percentile of the shuffle distribution). In the delay area, we did not identify turn-selective coding above chance level (percent significant cells, on/10 s: 5.2%; on/30 s: 5.7%; off/10 s: 4.7%; off/30 s: 5.7%; Fig. 7a and Fig. S8d). To confirm that our analysis of rate differences is in principle highly sensitive for detecting turn-selective coding, we also applied the method to cells that were active in the T zone (i.e., where the path towards the left and right had diverged) and found that 50-60% of cells were identified as selective for one or the other arm (Fig. 7b). Conversely, we tested whether trajectory coding would increase when the initial path segment in the delay zone, before the head was at a consistent position in front of the barrier, was included. Here, we found a significant proportion of turn-selective cells for the 10 s delay conditions (cells, on/10 s: $p = 0.002$; on/30 s: $p = 0.15$; off/10 s: $p = 4.1 \times 10^{-5}$; off/30 s: $p = 0.055$, $n = 206$). Finally, we tested whether turn-selective coding would be higher when selectively including only reactivation during population events, which occurred during the delay in the treadmill off conditions. We confirmed that most population events in the delay interval and at the reward locations were accompanied by sharp-wave ripples (SWRs; 54.5%/52.6% in delay/reward; Fig. 7c–e). However, we did not find turn-selective

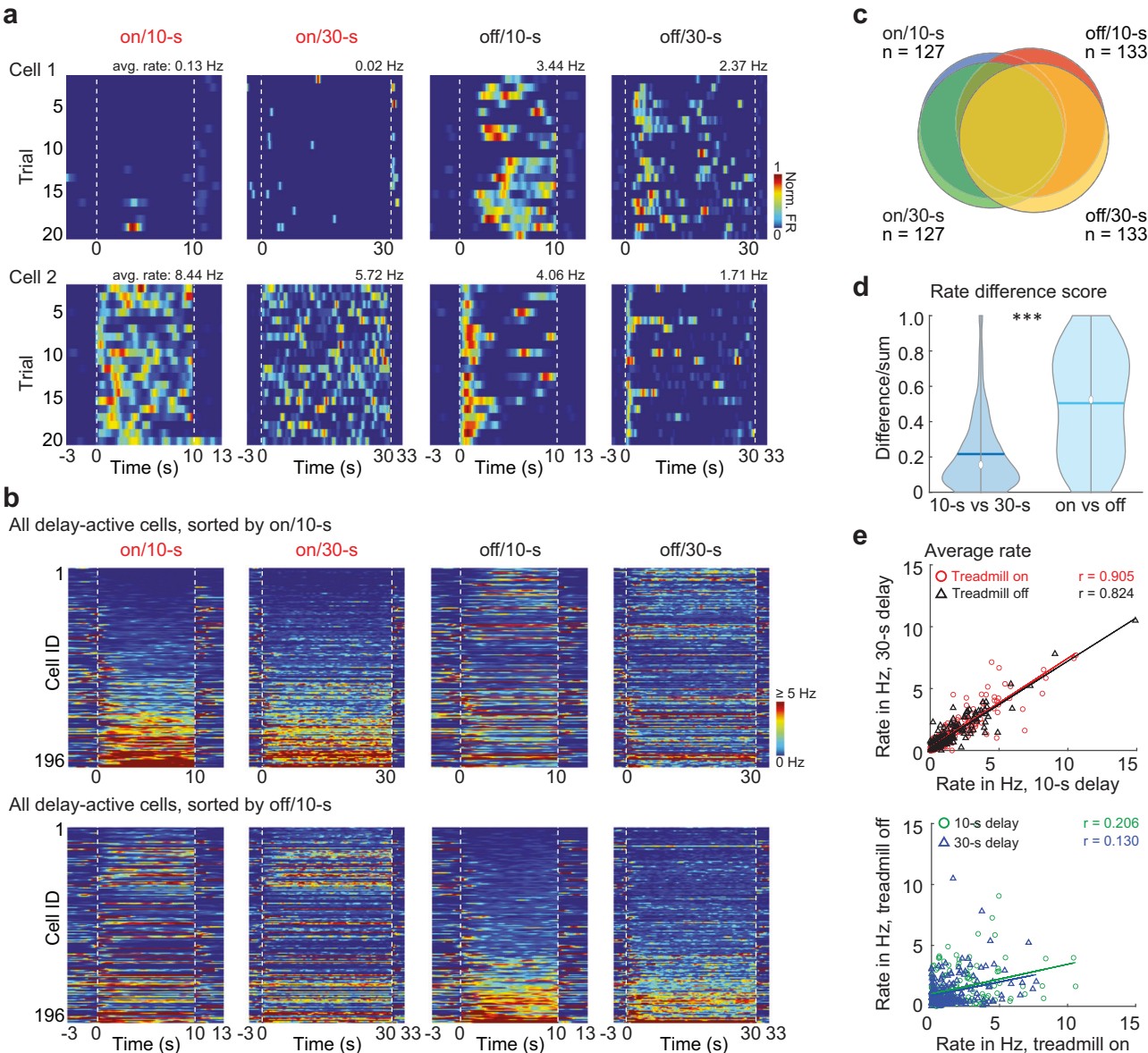

**Fig. 3 | Firing patterns of delay-active cells changed to a larger extent across running status rather than with delay length. a** Firing patterns of two delay-active cells across trials. One was active in only treadmill-off conditions (top) and the other in all conditions (bottom). Rate maps are scaled to the maximum firing rate (upper right corner, in Hz) for each condition. **b** Delay-active cells sorted by their average firing rate during the on/10 s delay condition (top) or their average firing rate during the off/10 s delay condition (bottom). **c** Overlap between delay-active cells across each of the four conditions (between on/10 s and on/30 s: 115; on10 s and off/10 s: 95; on/10 s and off/30 s: 84; off/10 s and off/30 s: 118; off/10 s and on/30 s: 91; on/30 s and off/30 s: 80). **d** Normalized firing rate differences

across delay/treadmill conditions (10 s vs. 30 s, mean ± SEM: 0.22 ± 0.01; on vs. off: 0.50 ± 0.02; $p = 1.0 \times 10^{-22}$, $n = 196$, Kolmogorov–Smirnov test; blue line, mean; white dot, median; gray outlines, distribution). **e** The average firing rates of delay active cells across 10-s and 30-s delay conditions were strongly correlated, in contrast to comparisons across the treadmill-on and off conditions (treadmill on: $r = 0.91$, $n = 196$, $p = 3.9 \times 10^{-74}$; off: $r = 0.82$, $n = 196$, $p = 1.2 \times 10^{-49}$; 10 s delay: $r = 0.21$, $n = 196$, $p = 0.004$; 30 s delay: $r = 0.13$, $n = 196$, $p = 0.09$; Spearman's correlation). All statistical tests are two-sided without post-hoc adjustments for multiple comparisons. **p < 0.05, ***p < 0.001. Source data are provided as a Source Data file.

reactivation above chance for population events in the delay zone (off/10-s: $p = 0.090$, $n = 92$; off/30 s: $p = 0.064$, $n = 194$). As a control, we analyzed left vs. right selective reactivation at the reward locations and detected clear differences (on/10 s: $p = 4.6 \times 10^{-13}$, $n = 86$; off/10 s: $p < 2.3 \times 10^{-308}$, $n = 139$; on/30 s: $p < 2.3 \times 10^{-308}$, $n = 92$; off/30 s: $p < 2.3 \times 10^{-308}$, $n = 86$; Fig. 7f). Finally, we performed analysis of similarity in average firing rates during the delay between correct and error trials and compared it to the baseline similarity between pairs of correct trials. The analysis revealed that firing rate differences to error trials were higher than the baseline in 10-s delays, but not in 30-s delays. This result was consistent across two datasets ($n = 5$ and 6 rats; Fig. S11a, b).

Given the lack of trajectory coding by single hippocampal cells in the delay area, we next asked whether trajectory information may be found in the activation of cell assemblies rather than of single cells. We focused on assemblies that were active during the delay interval, and using independent component analysis[28,29], identified 67 assemblies in the treadmill-on condition and 106 assemblies in the treadmill off-condition, which showed activity at various times throughout the delay interval (Fig. S11c, d, Fig. S12a, b). We next asked whether the assemblies that were identified during the treadmill-on and off conditions were the same or distinct by setting a threshold of 0.5 for the correlation of assembly weights to distinguish between common and distinct assemblies (Fig. S11e, f). As expected from the higher number

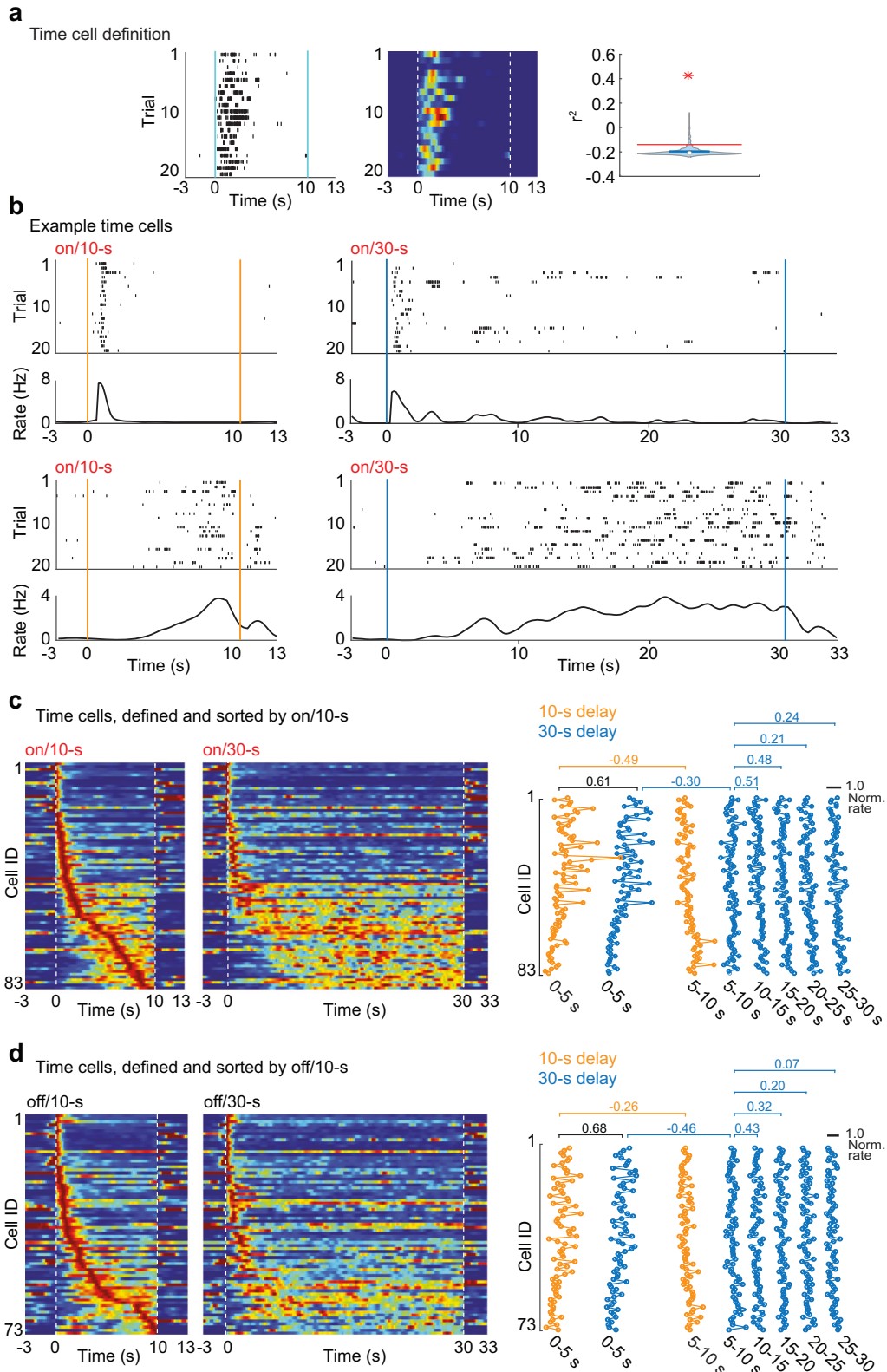

of assemblies during the treadmill-off condition, there were many unique assemblies that only occurred in off conditions. To the contrary, many of the assemblies in the treadmill-on condition were similar to those in the off condition, and the assemblies that were identified during the treadmill-on condition with ongoing theta oscillations showed a theta phase preference (Fig. S12c). Given that assemblies did not show complete overlap between conditions, we

performed the subsequent analysis of trajectory-dependent coding by separately using assemblies detected with treadmill on for the treadmill-on condition and assemblies detected with treadmill off for the treadmill-off condition. Despite the large pool of assemblies that was detected during each type of delay, assembly activation during the delay was not consistently informative for one or the other turn direction beyond chance levels (Fig. 7a and Table S4). We repeated the

**Fig. 4 | Two types of firing patterns were discernible during the delay interval. a** Unbiased classification of time cells. Cells with trial-to-trial stability greater than the 95th percentile of shuffled data were classified as time cells. Example time cell that was active during the first few seconds across trials (left: spike raster plot; center: corresponding firing rate map across trials; blue, 0 Hz; red, maximum firing rate). Right: calculations for the same cell (red star, trial-to-trial correlation; for shuffled data: red line, 95th percentile; blue line, mean; white dot, median; gray lines, distribution). **b** Time cells were divided into time-limited cells and persistently active cells, depending on whether they continued to be active until the end of the delay period. Raster plots of example cells representing each type are shown. Top: time-limited cell with activity early in the delay interval in both the 10 s delay (left) and 30 s delay (right) conditions. Bottom: persistently active cell that turned on after -5 s and remained active until the end of the 10 s delay (left) and 30-s delay (right). **c** Left: all time cells in the on/30 s condition, and their firing patterns in the corresponding order in the on/30 s condition. Each line is a cell, and cells were sorted by the time of peak firing in on/10 s. Blue, 0 Hz; red, each cell's maximum rate within a condition. Most time cells that were activate late in the 10 s delay were also persistently active in the 30 s delay. Right: for time cells identified in the on/10 s condition, normalized firing rates were calculated for every 5 s interval (by dividing each cell's rate in each 5 s interval by the cell's average rate over the 30 s delay).

Firing rates in the 0–5 s interval were highly correlated between the 10 s and 30 s delay conditions ($r^2 = 0.61$, Pearson's linear correlation). Firing rate patterns differed between the 0–5 s and 5–10 s intervals (on/10-s: $r^2 = -0.49$; on/30 s: $r^2 = -0.30$, Pearson's linear correlation), but then remained similar between the 5–10 s interval and the remainder of the 30 s delay (vs. 10–15 s: $r^2 = 0.51$; vs. 15–20 s: $r^2 = 0.48$; vs. 20–25 s: $r^2 = 0.21$; vs. 25–30 s: $r^2 = 0.24$, Pearson's linear correlations; see Table S1 for statistics). **d** Left: all time cells in the off/10 s condition, and their firing patterns in the corresponding order in the off/30 s condition. Each line is a cell, and cells were sorted by the time of peak firing in off/10 s. Most time cells that were active late in the 10 s delay were also persistently active in the 30 s delay. Right: for time cells identified in the off/10 s condition, normalized average firing rates were calculated for every 5 s interval throughout the delay. Firing rates in the 0–5 s interval were highly correlated between the 10 s and 30 s delay conditions ($r^2 = 0.68$, Pearson's linear correlation). Firing rate patterns differed between the 0–5 s and 5–10 s intervals (on/10-s: $r^2 = -0.26$; on/30-s: $r^2 = -0.46$, Pearson's linear correlation), but then remained similar between the 5–10 s interval and the remainder of the 30 s delay (vs. 10–15 s: $r^2 = 0.43$; vs. 15–20 s: $r^2 = 0.32$; vs. 20-25 s: $r^2 = 0.20$; vs. 25–30 s: $r^2 = 0.07$, Pearson's linear correlations; see Table S1 for statistics). All statistical tests are two-sided without post-hoc adjustments for multiple comparisons. Source data are provided as a Source Data file.

analysis for groups with only treadmill-on or only treadmill-off training. Only with treadmill-on training, assemblies carried information about turn direction in the 10 s delay, but not in the 30 s delay (Table S4). This suggests that running during the delay can render cell assemblies informative, but only over short durations.

Because we could not consistently detect turn-selective cells or cell assemblies during the delay period, we tested whether cells or cell assemblies may be selectively reactivated during the brief period on the stem, after the exit from the delay zone. We restricted the analysis to a segment on the center arm before the path diverged by >2 cm from the midline. With this criterion, we did not observe a significant proportion of turn-selective cells for any of the delay/treadmill conditions. Once the path starts diverging (50−60 cm from the delay zone, -2.5 cm to the left, 2.3 cm to the right), a significant fraction of cells was left vs right selective (Fig. 7g; 15.0% of 100 cells, $p = 0.001$, binomial test compared to chance level). We then examined whether cell assemblies would be informative in the stem segment before the path diverges. Although the rats only spend a few seconds in this zone in each trial, we found turn-selective activity of a high fraction of cell assemblies (Fig. 7g, Fig. S13 and S14). However, the turn-selective coding of assemblies was only found in conditions when the treadmill had been off during the preceding 10 s and 30 s delay periods.

## Discussion

Ongoing or sequential activity patterns during delay intervals of several seconds can support WM, but it is unclear for how long such a mechanism can be sustained and whether the length of sequences during the delay interval depends on the ongoing theta oscillations. To address these questions, we recorded from hippocampal cell populations during a hippocampus-dependent spatial WM task in which we varied delay length and manipulated predominant brain oscillation patterns during the delay. Differences in brain oscillations were achieved by either having rats rest during the delay interval or forcing them to run during the delay. We confirmed that our behavioral manipulation resulted in the expected differences in theta oscillations during the delay, with longer bouts and higher theta amplitudes during running compared to rest. Despite the major differences in oscillation patterns, we did not find differences in the extent of sequential activity patterns and found that sequential activity occurred predominantly during the first -5 s in the delay. After the initial few seconds, there was a qualitative change in hippocampal firing patterns, and a subpopulation of cells remained persistently active for the remainder of the delay interval. While the proportion of time-selective cells during the delay depended on the training history, these cells were not found to be informative about

either the past or upcoming turn direction. In addition, we analyzed coincident firing of subsets of principal cells. As expected, these events were more common during rest than during running in the delay, but in either case, not informative about turn direction during the delay. Rather, a large fraction of cell assemblies ( ~ 30%) was informative during the brief period immediately following the delay period, but only if rats were not forced to run during the preceding delay interval. Together, these data suggest that the hippocampus contributes to memory coding in spatial WM tasks by mechanisms other than generating sequential activity during the delay. Furthermore, given that coding in the stem following the delay is selective in only a subset of conditions (see Fig. 7g), it is also not a mechanism that generalizes to support all types of spatial WM.

Previous recordings from hippocampal cells in WM tasks have identified sequence or time cells over intervals up to 15 s[8,9,11], with sequential activity while theta oscillations were ongoing[18]. In studies that manipulated theta oscillations, sequential activity was disrupted along with reduced behavioral performance or was sustained along with intact performance[8,18,19]. As in previous studies, our data also identified time cells, which were precisely tuned to time early in the delay and broadly active late in the delay. Yet, these data allow for the alternative interpretation that there is a qualitative shift in hippocampal firing patterns at about 5 s into the delay intervals, when an early, sequentially active cell population is replaced by a later cell population with sustained activity throughout the remainder of the delay interval. Since late-active cells generally fire throughout most of the second half of the 10 s delay, their firing fields are broader, as previously reported[25]. Undoubtedly, it would have been difficult to distinguish between the interpretation of gradually broadening time fields and a qualitative shift from time cells to persistently active cells based on the 10 s delay data alone. However, our findings show a clear transition from sequentially active cells to persistently active cells with 30 s delay intervals. Sequential activity patterns thus have a limited lifetime and are not sustained across long delay intervals, even in conditions when theta oscillations are ongoing throughout the delay. However, it is feasible that the high running speeds in some previous studies[8,11] resulted in more vigorous theta engagement and extended the temporal span of sequential activity to intervals of -15 s. Irrespective of potential mechanisms that recruit time cells over longer intervals, we show that sequential firing patterns throughout the delay are not necessary for memory retention, including in conditions with uninterrupted theta oscillations. In our data, there were no sustained time cell sequences during the delay interval—with or without sustained theta—when memory was intact. Along with the finding that

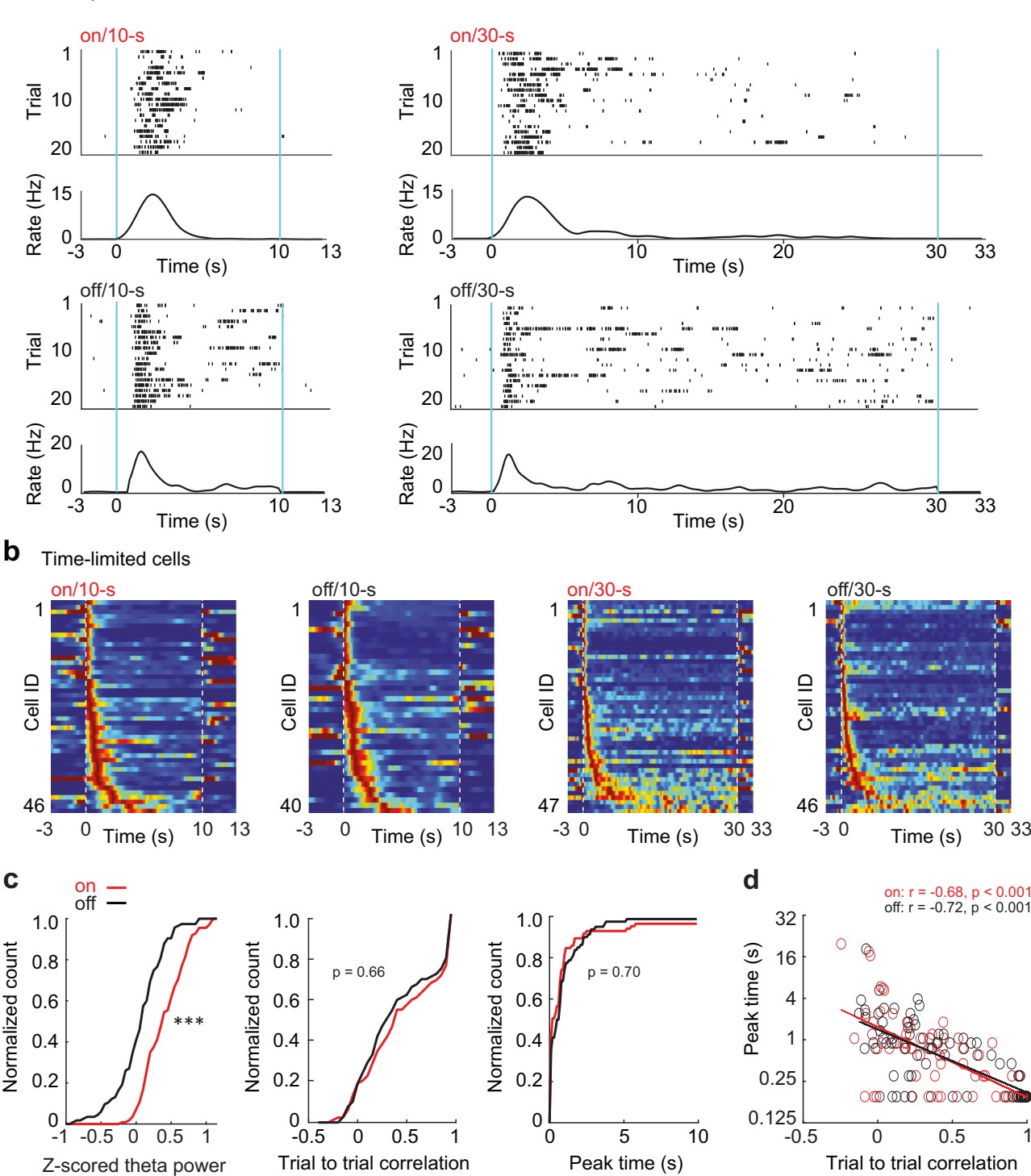

**Fig. 5 | Time-limited cells predominantly coded for the initial delay period.**
**a** Top: raster plots of example time-limited cell in treadmill-on delays, which had similar firing patterns in the 10 s delay (left) and 30 s delay (right). Bottom: raster plots of the example time-limited cell in the treadmill off delay intervals, again with similar firing patterns in both delay conditions. **b** Time-limited cells sorted by the peak time within each delay condition. Blue, 0 Hz: red, each cell's maximum firing rate within a condition. The peak firing rates of time-limited cells occurred predominantly in the first 5 s of the delay period regardless of the delay duration (on/10 s: 98.5% ± 1.6; off/10 s: 100.0% ± 0.0; on/30 s: 84.9% ± 7.92; off/30 s: 82.8% ± 8.54, mean ± SEM, $n$ = 18 sessions). **c** From left to right: z-scored theta, time field stability

and peak time distribution of time-limited cells. Despite the major difference in theta amplitude between treadmill on and off, time-field stability and peak times did not differ across conditions (z-scored theta: $p$ = 8.0 × 10⁻⁹; time field stability: $p$ = 0.66; peak time: $p$ = 0.70, Kolmogorov-Smirnov test). Distribution of peak times and time field stability were unrelated to theta power. **d** Higher time field stability (i.e., trial-to-trial correlation) was found for time-limited cells with earlier firing peaks (treadmill on: $r$ = -0.68, $n$ = 93, $p$ = 1.2 × 10⁻¹⁶; treadmill off: $r$ = -0.72, $n$ = 86, $p$ = 4.1 × 10⁻¹⁴, Spearman's correlation). All statistical tests are two-sided without post-hoc adjustments for multiple comparisons. *** $p$ < 0.001. Source data are provided as a Source Data file.

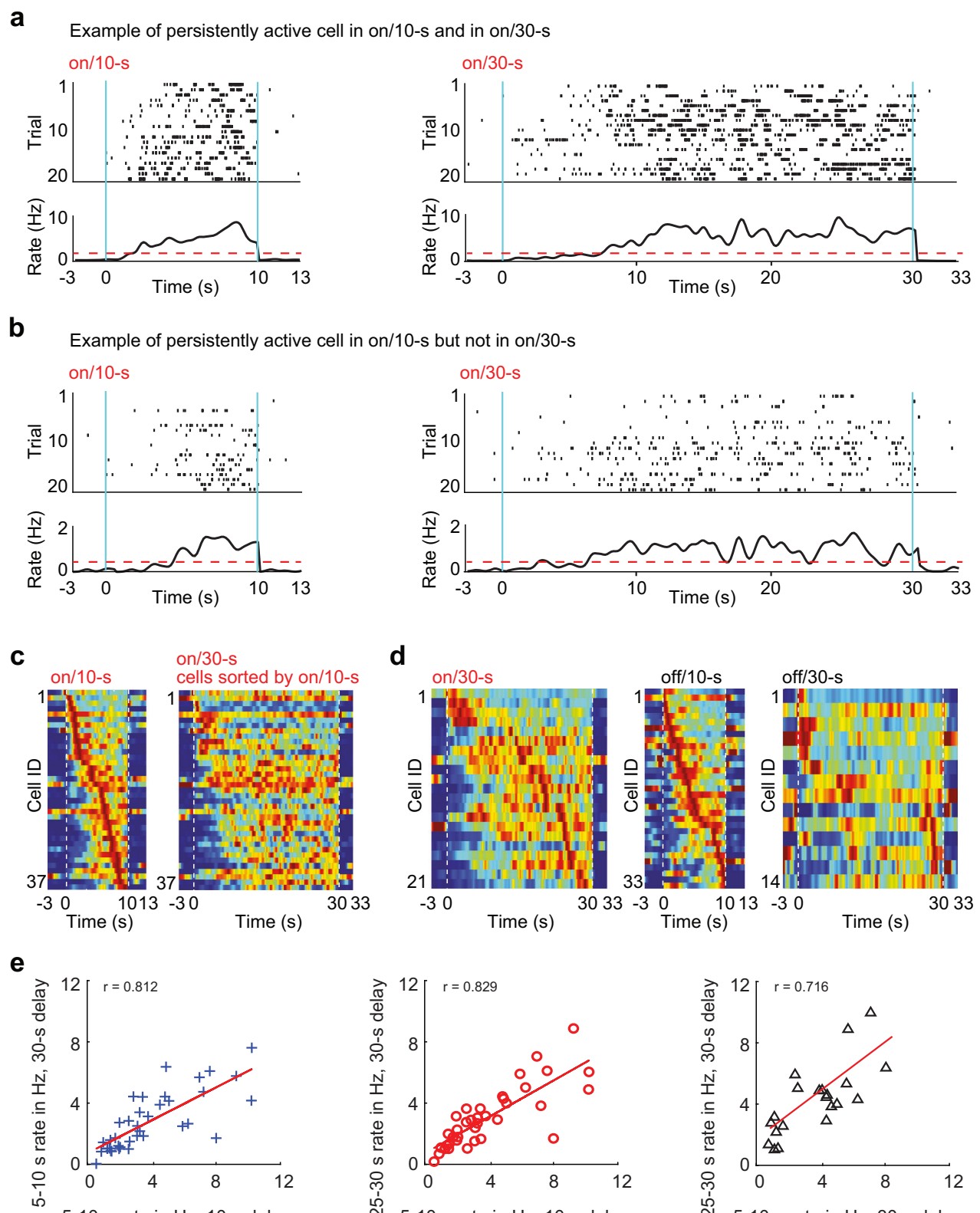

**a** Example of persistently active cell in on/10-s and in on/30-s

**b** Example of persistently active cell in on/10-s but not in on/30-s

within-cycle theta sequences can be preserved when spatial memory is impaired[30], there is thus no consistent correspondence between sequential firing patterns—either within the theta cycle or within the delay interval—and memory performance.

Relatedly, an additional unexpected finding in our data is that neither the early time cells nor the later persistently active cells were informative about turn direction, which is conflicting with some

previous reports[8,9], but not the only report of either a low or non-significant fraction of turn-selective cells in delay intervals without running[13,16,20]. Here, we directly tested whether running and its asso-ciated continuity of theta oscillations are a possible source for the discrepancy between studies with and without prolonged sequential firing patterns[8,9,16] by including blocks of trials with and without run-ning in our experiment. There were no major differences in the

**Fig. 6 | Persistently active cells remain active until the end of the delay period. a** Time cells were classified as persistently active cells when their activity persisted until the end of the delay period. Raster plot of an example cell that was persistently active in the treadmill-on condition during both delay intervals (left: 10 s delay; right: 30 s delay). **b** Raster plot of an example cell that was persistently active cell in the treadmill-on/10 s delay (left), but failed to classify as persistently active cell in the treadmill-on/30 s delay condition (right). **c** All persistently active cells, identified during the on/10 s delay condition and sorted by their peak firing times (left). The activity patterns for the same cells are shown in the corresponding order for the on/30 s delay condition (right). Blue, 0 Hz; red, each cell's maximum rate within a condition. **d** Left to right: persistently active cells identified in the on/30 s delay, off/10 s delay and off/30 s delay conditions, sorted by their peak firing within each condition. Blue, 0 Hz; red, each cell's maximum rate within a condition. **e** Left: for the treadmill-on condition, the average firing rates of persistently active cells in the 5-10 s interval were strongly correlated across the two different delay lengths ($r = 0.812$, $n = 37$ cells, $p = 1.1 \times 10^{-7}$, Spearman's correlation). Center and right: firing rates in the 5–10 s interval in the 10 s delay and in the 30 s delay were strongly correlated with firing rate in the 25–30 s interval in the 30 s delay (25–30 s; $r = 0.83$, $n = 37$ cells, $p = 5.5 \times 10^{-8}$; $r = 0.72$, $n = 21$ cells, $p = 3.9 \times 10^{-4}$, Spearman's correlation). All statistical tests are two-sided without post-hoc adjustments for multiple comparisons. Source data are provided as a Source Data file.

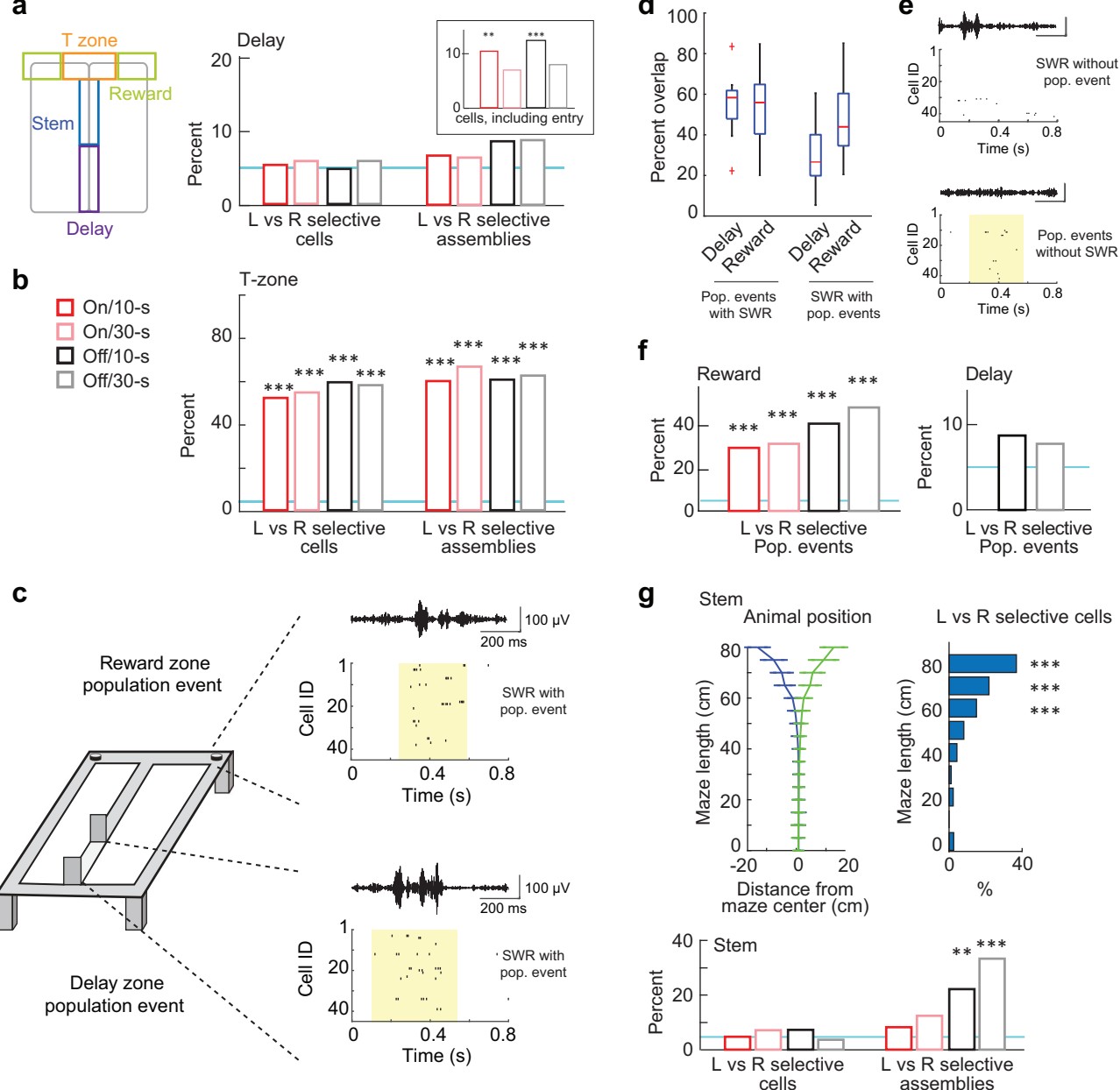

fraction of time cells or in the length of the time period with sequential firing patterns across these conditions. This raises the question whether other differences across studies could explain the discrepancy. First, we tested whether the criterion for identifying time cells could influence the result, which was not the case. Second, we tested whether the extent of familiarity leads to more memory-related firing. We found a higher proportion of time cells with less training, but time cells were nonetheless not turn-selective. Third, we included the initial entry path into the delay zone in the analysis and found an increase in the proportion of turn-selective cells. While this is perhaps not surprising, considering that the path had not converged to a common behavioral pattern across right-turn and left-turn trials, it is consistent with the

**Fig. 7 | The activity of hippocampal time cells and cell assemblies coded for left-turn vs right-turn trials after exiting the delay zone, but not during the delay.** **a** Left, schematic of defined maze regions. Right, proportion of turn-selective cells or cell assemblies in the delay zone. No significant turn selectivity was observed during the delay period (cells: on/10 s, $p = 0.50$; on/30 s, $p = 0.37$; off/10 s, $p = 0.50$; off/30-s, $p = 0.37$, $n = 196$ cells; assemblies: on/10 s, $p = 0.36$, $n = 61$; on/30 s, $p = 0.40$, $n = 64$; off/10 s, $p = 0.10$, $n = 95$; off/30 s, $p = 0.08$, $n = 105$ assemblies; chance level = 5%, blue dotted line, binomial tests) except when the analysis segment was expanded to include the entry into the delay zone (i.e., with the head, but not the body inside the delay zone; inset: cells, on/10 s, $p = 0.002$; on/30 s, $p = 0.15$; off/10 s, $p = 4.1 \times 10^{-5}$; off/30 s, $p = 0.055$, $n = 206$ cells, binomial tests). **b** Proportion of turn-selective cells ($p < 2.3 \times 10^{-308}$, $n = 148$ cells for all treadmill/delay conditions, binomial tests) and turn-selective assemblies (on/10 s, $p = 2.7 \times 10^{-6}$, $n = 10$; on/30-s, $p = 2.3 \times 10^{-6}$, $n = 15$; off/10-s, $p = 1.0 \times 10^{-17}$, $n = 28$; off/30-s, $p < 2.3 \times 10^{-308}$, $n = 32$ assemblies; chance level = 5%, blue dotted line, binomial tests) in the T zone. **c** Two example population events detected in the reward area and during the delay in the treadmill-off condition. Both CA1 population events (spike rasters with yellow background) occurred concurrently with SWRs (CA1 LFP trace on top of spike rasters, filtered at 150–250 Hz). **d** Overlap between population events and SWRs. Left: proportion of identified CA1 population events that were associated with SWR (treadmill-off delay: 54.5% ± 4.25; reward area: 52.6% ± 5.67, $n = 12$ sessions). Right: the proportion of detected SWRs in the CA1 LFP associated with CA1 population events (treadmill-off delay: 29.5% ± 4.64; reward area: 48.8% ± 5.76, $n = 12$ sessions). **e** Top, example SWR that did not co-occur with a population event. Bottom,

example population event that did not co-occur with a SWR. **f** Population event-associated firing rates were significantly turn-selective in the reward area (left; on/10-s, $p = 4.6 \times 10^{-13}$, $n = 86$; off/10-s, $p < 2.3 \times 10^{-308}$, $n = 139$; on/30 s, $p < 2.3 \times 10^{-308}$, $n = 92$; off/30-s, $p < 2.3 \times 10^{-308}$, $n = 86$ population events, binomial tests), but not during delay intervals (right; off/10 s, $p = 0.090$, $n = 92$; off/30-s, $p = 0.064$, $n = 194$ population events; chance = 5%, blue dotted line, binomial tests). Note that only treadmill-off, but not treadmill-on trials had a sufficient number of population events during the delay for analysis of turn selectivity. **g** CA1 cell assemblies, but not single CA1 cells were significantly turn-selective when rats traversed in the maze stem. Top left, average head position along the maze stem (5 cm bins, aligned to the running direction) during left- and right-turn trials (mean ± SD, $n = 18$ sessions). Top right, proportion of turn-selective cells per spatial bin along the stem (10 cm bins; 0 cm, $p = 0.24$, $n = 79$; 10 cm, $p = 1.0$, $n = 88$; 20 cm: $p = 0.18$, $n = 88$; 30 cm, $p = 0.07$, $n = 85$; 40 cm, $p = 0.50$, $n = 93$; 50 cm, $p = 0.12$, $n = 99$; 60 cm, $p = 0.0001$, $n = 15$; 70 cm, $p = 1.8 \times 10^{-10}$, $n = 119$; 80 cm, $p < 2.3 \times 10^{-308}$, $n = 114$, binomial tests). Bottom, significant turn selectivity was observed specifically for cell assemblies in treadmill-off trials (cells: on/10 s, $p = 0.51$; on/30 s, $p = 0.11$; off/10-s, $p = 0.11$; off/30 s, $p = 0.27$, $n = 211$; assemblies: on/10 s: $p = 0.34$, $n = 24$; on/30-s: $p = 0.12$, $n = 24$; off/10 s: $p = 0.002$, $n = 27$; off/30-s: $p = 1.2 \times 10^{-6}$, $n = 30$; chance = 5%, blue dotted line). All statistical tests are two-sided without post-hoc adjustments for multiple comparisons. Box plots: central line, edges, whiskers and plus signs indicate median, the 25th/75th percentile, maximum/minimum and outliers. *$p < 0.05$, **$p < 0.01$, ***$p < 0.001$. Source data are provided as a Source Data file.

finding that most differentially active cells are found early in the delay period[8]. Finally, our training protocol included a phase with continuous spatial alternation, and such training could perhaps reduce differential firing later in the task. However, turn-selective firing is particularly abundant in the continuous task version[27], which makes it unlikely that training in this task variant occludes turn-selective firing. Similarly, a task variant with a high proportion of turn-selective cells requires running back and forth in opposite directions along a common path. This behavioral pattern resembles running back and forth on a linear track, which is known to result in direction-selective firing[31,32] that may extend to the included running wheel[18]. Although there are thus examples of experimental conditions in which two separate hippocampal firing patterns emerge across choices for which WM is required[8,33–35], these findings do not generalize. Sequential and memory-related hippocampal firing patterns during delay intervals are not consistently found in hippocampus-dependent WM tasks, including in the large sample in our study (n = 19 rats). In particular, sustained theta oscillations during the delay interval do not necessarily result in memory-related firing patterns during the delay.

If the sequential firing of hippocampal cells throughout the delay interval is not informative, but behavior is hippocampus dependent[8,13], what are other mechanisms by which hippocampal neuronal activity can contribute to memory? It is feasible that the hippocampus contributes to WM performance by providing spatial information to other brain regions before the delay. This is supported by a role of ventral hippocampus during the sampling phase of spatial WM tasks and by the finding that manipulations of hippocampal and entorhinal theta oscillations are effective when targeted to the return arms of the maze[21,36]. During trace eyeblink conditioning and for spatial memory, either the average hippocampal activity throughout the delay or brief activity patterns during SWRs have been shown to be informative about upcoming choices[35,37]. We therefore analyzed whether the cumulative firing rate of individual cells or SWR-related population activity during the delay were memory-related in our task. First, we did not find information beyond chance levels in the firing rates of individual cells. Second, we analyzed cell assemblies and found a high number of distinct assemblies in the delay zone. While assembly activation was not informative when occurring during the delay, we found that their reactivation on the stem—after delay periods without running—was highly predictive of turn direction. Similar findings have previously also been reported for single cells[20], but in our single cell

analysis, the effect was less pronounced (see Fig. 7g). For some types of delay intervals, we therefore found a possible hippocampal contribution to re-instantiating memory-related firing patterns. However, the finding that such memory-related cell assembly re-activation occurred predominantly after delay periods without persistent theta suggests that assembly organization on the stem depends on the oscillatory dynamics during the preceding delay interval. For example, it can be speculated that short-term plasticity—either locally or in a larger brain circuit—occurs in treadmill-off trials, which later initiates turn-selective activity of hippocampal assemblies. If these mechanisms specifically occur outside of theta oscillations (e.g., SWRs/population events), the same computations would not be engaged in treadmill-on trials, which then require a different set of computations to support memory retention. While we thus found only limited information about turn-direction, we identified higher trial-by-trial inconsistency in hippocampal firing rates during 10-s delay intervals for error compared to correct trials. While the hippocampal activity during the delay may therefore not be informative about turn-direction, consistent firing patterns across trials (see Fig. S11a, b) are—for short delay durations—indicative of upcoming correct choices.

Taken together, our findings confirm that sequential activity patterns during the initial seconds in a delay interval are a widely found characteristic of hippocampal population activity. However, sequential activity patterns were not prolonged by ongoing theta oscillations, and irrespective of the extent of theta oscillations during the delay interval, were followed by persistent activity for the remainder of the delay interval in a subpopulation of hippocampal cells. As expected, we observed that hippocampal oscillation patterns, such as theta oscillations and SWRs, depended on the behavioral state during the delay interval (i.e., running or resting). Irrespective of the occurrence of either theta or SWRs, there were large similarities in the fractions of sequentially and persistently active cells without consistent evidence for memory-related activity patterns during any of the delay conditions. Our study thus suggests that sequentially active hippocampal firing patterns are short-lasting and not a universal mechanism for sustaining WM.

## Methods
### Approvals
All experimental procedures were conducted at the University of California, San Diego according to National Institutes of Health

guidelines and were approved by the Institutional Animal Care and Use Committee.

## Animal subjects and behavioral task

The experiments used 26 Long-Evans rats (14 males and 12 females), which were housed individually on a reverse 12 h light/dark cycle. During behavioral training and recording, rats were food restricted to around 85% of the *ad libitum* weight. The behavior apparatus was a white figure-eight shaped maze (150 cm x 100 cm) constructed from white plastic sheets. All sections of the figure-eight maze were composed of 10 cm wide tracks with 2 cm tall ridges on each side. The delay area (70 cm long, 10 cm wide) was located in the half of the center arm that the rat first entered. A treadmill (Harvard apparatus) was fitted into the delay area as the floor for the rats to rest or run on (see Fig. 1). During the delay periods, the rats were spatially restrained in the delay area by an automatic barrier at the front of the delay zone and by a manual barrier behind its tail. The manual barrier was placed behind the rats after they entered the delay area to prevent them from exiting the delay area.

We trained 6–10 weeks old rats to collect chocolate sprinkles at two corners of the figure-eight maze (i.e., the reward locations) with food only available at the reward location if the rats chose the opposite turn compared to the previous trial. During the initial pre-training phase, the rats were given 5 days to habituate and explore the maze freely. Next, they were trained to perform a continuous version of the alternation task for 3–5 weeks until they reached ≥90% accuracy in two out of three consecutive training days. Then, a delayed version of the alternation task was introduced to all rats, and their training continued for 10 days before electrode implantation. Eighteen rats (10 male and 8 female) were trained with four trial types by combining two delay interval durations (10 s or 30 s) with the two treadmill states (off or on) during the delay. Ten consecutive trials with each of the four trial types were performed with the order of the blocks of trial types shuffled across recording days. Within each training/recording day, the four blocks were repeated for a second time in the same order (see Fig. 1). In addition to the experimental group in which the training included blocks with the treadmill on and off, we trained two additional groups of rats (*n* = 4 rats per group, 2 male and 2 female rats). For one of the groups, the treadmill was always on and for the other group, the treadmill was always off. With the two interval durations (10 s and 30 s), this resulted in two conditions per group, and ten consecutive trials were performed with each of the two conditions with the order of the conditions shuffled across recording days. Within each training/recording day, the two conditions were repeated for a second time in the same order. Training and recording sessions were otherwise performed as described for the group with combined training. In addition, recording sessions included 30-min sleep blocks before and after the alternation task.

## Surgery and recording methods

**Hyperdrive.** Fifteen pretrained rats were selected for hyperdrive implantation (*n* = 7 with combined treadmill-on and off training, *n* = 4 with only treadmill-on training and *n* = 4 with only treadmill-off training). Twelve moveable tetrodes (90% platinum/10% iridium wire) and 2 moveable reference electrodes were assembled into a hyperdrive, which was implanted after the initial training phase (i.e., when the trained rats were 4–7 months old). Anesthesia was induced with isoflurane (2–3% mixture in oxygen, 1 L/min), and buprenorphine (1 mg/kg) was administered as an analgesic. After placing skull screws and performing a craniotomy (4.0 mm posterior to the bregma and 2.7 mm to the right of the midline), the hyperdrive was secured with dental cement. We began to lower tetrodes at the end of surgery.

Rats were allowed to recover for 5 days in their home cage with unlimited access to food and water. After the recovery period, they were again food restricted and re-trained on the maze until they ran 80

non-delayed trials (combined treadmill-on and off training group) or 40 non-delayed trials (only treadmill-on or only treadmill-off training groups) in the alternation task. On average, it took ~5 days for the animals to be retrained with non-delayed trials after surgery, and training with the delays was then resumed. During this period, tetrodes were slowly advanced toward the hippocampus and placed into the hippocampal CA1 area. Recordings began when stable signals were obtained from CA1 and when subjects ran consistently in the figure-eight maze task. Using the Digital Lynx data acquisition system running the Cheetah software (Neuralynx, Bozeman, MT, USA), unit recordings were obtained by amplifying and filtering the signal with a 600−6000 Hz bandpass filter and by digitizing 1 ms-long segments at 32 kHz when a spike exceeded a threshold of 45−50 μV. In addition, LFP from one channel of each tetrode was acquired by continuously recording with a lowpass filter at 450 Hz and with a sampling frequency of 2 kHz. Data from two rats (1 male, 1 female) in the group with combined treadmill-on and off training were excluded due to poor recording quality. Of the 7 trained rats in the combined training group, 5 were therefore used for analysis.

**Neuropixels.** Neuropixels 1.0 probes (IMEC)[38] were chronically implanted in eleven pretrained rats with combined treadmill-on and off training by following the same general surgical procedures as for hyperdrive implants. The use of a metal drive and protection case for the Neuropixels implants was adopted from the design described in a previous study[39]. The probe targeted the dorsal and ventral hippocampus with an entry coordinate of 3.0–3.3 mm posterior to the bregma and 1.5–1.7 mm lateral to the midline. The probe was tilted 30–35 degrees along both the anteroposterior and mediolateral axes and advanced 7–9 mm from the brain surface.

Rats were given 4–5 days to recover in their home cages with unlimited access to food and water. Following recovery, they were food-restricted and resumed the delayed spatial alternation task for recordings. The action potential (AP) band signals were recorded with a gain of 500 and a sampling frequency of 30,000 Hz, while the LFP band signals of matching AP channels were recorded with a gain of 250 and a sampling frequency of 2,500 Hz. The reference was connected to the ground using a silver wire (AGW1010, World Precision Instruments) attached to a ground screw implanted above the cortex. Data acquisition was performed using SpikeGLX (version 20201103) via the PXIe acquisition module card (IMEC) installed in the National Instruments PXIe chassis (PXIe-1071, National Instruments). Positional data were recorded using the Digital Lynx data acquisition system (Neuralynx, Bozeman, MT, USA). To synchronize positional and electrophysiological data, an Arduino sent a 0.5 Hz TTL signal to both the Neuropixels acquisition system and the Digital Lynx data acquisition system. Data from five rats (3 male, 2 female) were excluded due to a low number of cells in the dorsal hippocampus. Of the 11 trained rats with Neuropixels implants, 6 were therefore used for analysis.

## Histology, tetrode tracking and cell sorting

Following the completion of all recordings, the animals were euthanized with sodium pentobarbital. The brains were fixed by transcardial perfusion using phosphate-buffered saline (1x PBS, 0.1 M, pH 7.4) followed by 4% paraformaldehyde in 0.1 M PBS. After additional fixation in 4% paraformaldehyde, the brains were cryoprotected in a 30% sucrose solution. Coronal sections (40 μm thickness) were obtained and mounted on glass slides.

For tetrode tracking in hyperdrive-implanted rats, the sections were stained with cresyl violet, and transmitted light images were taken with a Leica CTR 6000 microscope to identify all electrode tracks. Tetrodes located in the hippocampal CA1 area were included in further analysis. Cluster cutting was performed manually by checking the spike distribution using a customized version of mClust[40]. Clusters

were classified as principal cells if their average firing rates were between 0.1 and 5 Hz.

Before surgery, Neuropixels probes were covered with 2% DiI (D282, Thermo Fisher) in pure isopropyl alcohol for probe tracing. The brains were processed as for hyperdrive-implanted rats. After coronal sections were mounted on glass slides, DAPI was applied before placing a cover glass. The images were taken with an emission wavelength of 455 nm for DAPI and 565 nm for the DiI using an Olympus slide scanner (VS200, Olympus). Cluster sorting was performed with Kilosort 2.0[41] (https://github.com/MouseLand/Kilosort), with the parameters as: ops.minfr_goodchannels = 0.01, ops.Th = [10  4], ops.minFR = 1/1000. Manual curation was done with Phy (https://github.com/cortex-lab/phy). Clusters were classified as principal cells if their average firing rates were between 0.1 and 5 Hz. All subsequent data analyses were performed using MATLAB.

## Local field potential analysis

To analyze theta oscillations during each recording session, one LFP channel was selected among the tetrodes which were confirmed to be located in the dorsal CA1 cell layer based on high cell numbers. To detect theta episodes, the LFP signal from whole recording sessions, including 2 sleep sessions and 80 trials of the alternation task, was band-pass filtered between 6–12 Hz. The envelope of the filtered signal was determined by Hilbert transform. Theta episodes were defined as periods when the envelope amplitude exceeded 0.5 standard deviations of the mean for at least 0.5 s. Two theta episodes were joined if their gap was ≤0.5 s. The initial theta bout length in the delay was determined by selecting the first theta episode end time in the delay zone, or if there was no theta detected when the animal entered the delay zone, by setting the initial theta bout length to 0 s.

## Definitions of delay-active cells, time cells, time-limited cells and persistently active cells

Spike trains obtained from single neurons were aligned by using the time when the animal reached the delay barrier (shown in Fig. 1) as the delay onset. If the average firing rate in the delay area was higher than 0.5 Hz in one of the four delay/treadmill conditions, a neuron was considered a delay-active cell. The temporal firing profile in the delay area was calculated by binning spikes into 150 ms windows followed by a Gaussian kernel convolution with a sigma of 300 ms.

For each neuron that had spikes in the delay area in >10 trials of any of the four conditions (from 20 trials per condition, obtained by combining the two 10-trial blocks in identical conditions), temporal stability was assessed by first calculating the Pearson's correlation between every pair of trials with spikes (10–20 trials resulted in 45–190 correlation values) and by then taking the median of all correlation values. A shuffled stability distribution was calculated for each neuron by randomly shifting spike trains in the delay interval for one of the trials in each pair (by at least 500 ms and with circularly wrapping the values). As for the actual data, the median of all pairwise comparison was then taken. The shuffling procedure was repeated 1000 times, resulting in 1000 shuffled values for each cell. If the actual stability was in the top 5th percentile of the shuffled stability distribution, the cell was considered a time cell in this delay/treadmill condition. If the firing rate of a time cell dropped to <20% of the peak rate before the end of the delay, it was classified as a time-limited cell. Otherwise, it was taken as a persistently active cell. Time fields were defined as the number of consecutive 150 ms bins above 20% of the peak firing time bin. For delay intervals in the treadmill-on condition, the classification of time cells, time-limited cells and persistently active cells was also done with only trials when theta bouts spanned >80% of the delay time.

## Theta precession of time cells

For spiking during delay intervals, theta phase precession within each cell's time field was measured. LFPs were filtered in the theta frequency band (6–12 Hz), and the instantaneous phase was extracted using the Hilbert transform. Spikes from time fields were assigned a corresponding theta phase. The circular linear correlation was computed between the spike theta phase and the normalized time field (start time as 0, end time as 1). A significant correlation along with a negative slope indicated theta phase precession[22].

## Assembly analysis

Cell assemblies were identified by using a measurement for coactivation of putative principal neurons[29] during the delay periods. Delay periods from trials with the treadmill either on or off were analyzed separately. Spike trains from each putative principal neuron were binned into 25 ms windows and were normalized with a Z-score transformation to form a spike matrix, and a correlation matrix was calculated for all neuron pairs in the spike matrix. The assembly number to be detected was determined by finding the significant eigenvalues of the correlation matrix which were larger than the threshold estimated by the Marčenko-Pastur law as a null hypothesis. Next, each assembly pattern was defined by the weight vector of neurons belonging to the patterns using the independent component analysis (ICA). For a cell to be included into the assembly, its weight needed to pass a threshold of $\frac{1}{\sqrt{n}}.\bar{w}$, where n is the number of all putative principal neurons and w is the assembly neuron weight from the ICA analysis.

The strength of the assembly pattern activity at a given time bin was reconstructed as:

$$R_k(t) = \mathbf{z}(t)^\mathsf{T}\mathbf{W}_k\mathbf{W}_k^\mathsf{T}\mathbf{z}(t) \tag{1}$$

where $R_k(t)$ is the strength of assembly k at time t, z is the spike count of each neuron, and w is the assembly neuron weight from the ICA analysis[28]. Assembly events correspond to instances when the R value (i.e., assembly strength) was higher than one standard deviation above the mean[42]. For each assembly detected in one type of delay interval (either treadmill on or treadmill off), the cosine similarity to the assemblies from the other type of delay interval was calculated, and the pair with the highest similarity value was considered the best match.

## Population activity and SWRs

During trials in the alternation task, rats remained stationary in the reward areas for short periods, and in treadmill-off trials, in the delay area. SWRs and bursts with co-active cells were observed during these recording segments. SWRs were identified using LFP recordings from a channel confirmed to be in the hippocampal CA1 layer based on histological verification. The signal was bandpass filtered between 150 and 250 Hz. SWR events were defined as periods when the filtered signal power exceeded three standard deviations above the mean for at least 50 ms. To further analyze the temporal activation pattern of principal neurons in the delay and reward areas, we selected recording sessions with at least 20 recorded principal cells and identified population events. A population event was defined as a 200 ms time bin when >1/5 of all neurons became active, but with neurons only included in the count if they were silent during the preceding 100 ms window (spike count ≤ 1). We reasoned that a criterion that directly selects brief periods of heightened neuronal activity, as otherwise also characteristic for SWRs, is best suited to analyze neuronal activity patterns. The population event onset was the first spike time of any of the neurons that were active in the event. If at least 20 population events were detected in the delay or reward area, the recording session was included for further analysis of population activity. In total, 12 recording sessions from 4 of the 5 rats met the analysis threshold.

## Turn-selective cells, assemblies and population events

For each delay condition, a difference score was calculated for each principal neuron and for each cell assembly to compare activity in left-turn vs. right-turn trials:

$$S_k(t) = (L_k(t) - R_k(t))/(L_k(t) + R_k(t)) \tag{2}$$

where $S_k(t)$ is the normalized rate/strength difference between left and right turn for neuron/assembly k at the maze area t, and L and R are the average firing rate/average assembly strength for left-turn and right-turn trials. The shuffled left vs. right rate/strength difference was calculated in the same way with turn directions randomly permutated 1000 times. If the original $S_k(t)$ was in the top 5th percentile of the shuffled S distribution, the cell or assembly was considered turn selective in the maze area t. Maze areas that were considered are the delay zone, the stem and the T zone (see Fig. 7a). Cells were excluded from the calculations if their average rate in the maze region under consideration was <0.5 Hz. Assemblies were excluded if they did not reach the activation threshold (i.e., one standard deviation above the mean) in the maze region under consideration in at least 4 of 20 trials. A similar analysis was applied to single-neuron spikes during population events in the treadmill-off delay intervals and the reward areas. To assess whether single-neuron activity during population events was associated with the replay of past trajectory information, this analysis was restricted to neurons that were active and were selective for left versus right return arms.

## Correct and error trial rate differences

To evaluate whether fluctuations of firing rates of single neurons during the delay period might contribute to incorrect choices, trial-by-trial rate differences between correct and error trials were calculated and compared to rate differences across pairs of correct trials. Specifically, for a delay condition with m error trials and n correct trials, rate differences were computed for each cell by first calculating

$$R_{diff} = abs(R_{error} - R_{correct})/(R_{error} + R_{correct}) \tag{3}$$

for all m × n error trial-correct trial pairs and by then taking the average over the cell's $R_{diff}$ values. To ensure that a sufficient number of error trial-correct trial pairs was available for averaging, only treadmill/delay conditions with at least 4 errors in 20 trials were included. As a baseline for comparison to the error-correct pairs, the same number of trial pairs (m × n) was randomly selected from all possible correct-correct trial pairs, and their average $R_{diff}$ was calculated.

## Statistical analysis

One-way ANOVA was used to analyze the behavioral performance (Fig. 1d), theta across trial types (Fig. 1g-j), active cell proportions on the maze and in the delay area (Fig. 2a), time-limited cell proportions (Fig. S6b), different cell proportions (Table S2 and S3) and time cell proportions with theta phase precession (Fig. S10a). Two-way ANOVA was used to compare different cell proportions from treadmill-on only and treadmill-off only training groups to those from the combined treadmill-on and off training group (Table S2 and Fig. S6b), methods for calculating time cell proportions (Fig. S8a) and proportions between Neuropixels and tetrode recordings (Fig. S8b). Paired t-tests were used to compare the speed (Fig. 1b) and theta power (Fig. 1f). Chi-square tests were performed to compare the fractions of delay-active cells across different delay types (Fig. 2a). Kolmogorov-Smirnov tests were used to compare rate difference scores (Fig. 3d), the distribution difference of time field stability, z-scored theta and peak time between treadmill-on and off blocks (Fig. 5c) as well as the similarity of assemblies across delay conditions (Fig. S11f). Spearman's correlations were used to compare the average rate between treadmill on vs off, delay 10 s vs 30 s (Fig. 3e), the relationship between time field stability

and field peak time (Fig. 5d), the relationship between 5 s blocks for persistently active cells (Fig. 6e), the average rate comparison between 10 s and 30 s delay in treadmill-on only and off only training groups (Fig. S7b) and peak time correlations between 10 s and 30 s delay intervals (Fig. S6d). Average rate distributions across 5 s segments were compared with Pearson's correlations (Fig. 4c, d). Binomial tests were used to compare turn-selective single cell and assembly proportions to the 5% chance level (Fig. 7a, b, f, and g). T-tests were used to compare the firing rate differences between error-correct trial pairs and pairs of correct trials (Fig. S11a, b). Average values correspond to the mean ± SEM unless indicated otherwise. The statistical significance was set to a level of 0.05, divided among two sides, except that binomial tests were one-sided. Post-hoc adjustments for multiple comparisons were not performed.

## Reporting summary

Further information on research design is available in the Nature Portfolio Reporting Summary linked to this article.

## Data availability

All the data reported in this paper will be shared by the corresponding authors upon request (sleutgeb@ucsd.edu, jleutgeb@ucsd.edu). Data can only be shared upon request because of their large size and because they are not in a standard format. Source data are provided with this paper.

## Code availability

All custom code for processing the data is freely available from a Github repository (https://github.com/LiYuan-001/Time-cell-sequences-during-delay-intervals-are-not-dependent-on-brain-states)[43]. Any additional information required to reanalyze the data reported in this paper is available from the corresponding authors upon request.

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

## Acknowledgements

We thank Crystal Hsu, Maylin Fu, Marina Jaramillo, and Mia Anderson for technical assistance. This work was funded by NIH grants R01 NS084324, R01 NS102915, R01 NS097772 to S.L. and R01 MH119179 to J.K.L and the Heiligenberg Professorship to J.K.L.

## Author contributions

L.Y., J.F.F., A.K., G.N. conducted the experiments, L.Y. performed the analysis, L.Y., J.K.L. and S.L. designed the experiments and L.Y., J.K.L. and S.L. wrote the paper.

## Competing interests

The authors declare no competing interests.
