## [Transparent Peer Review file · Nature Communications]

Time cell sequences during delay intervals are not dependent on brain state and do not support hippocampus-dependent working memory

Corresponding Author: Dr Stefan Leutgeb

Version 0:

Reviewer comments:

Reviewer #1

(Remarks to the Author)

The authors investigate whether time cells can be active outside of periods of theta activity. To this end, they propose a task where rats are forced to run or rest (run periods associated with theta oscillations). They found that time cells mainly represented the first few seconds of each delay period and that these neurons were not related to working memory. Populations of time cells were also state-specific and less affected by the duration of the delay. They conclude that working memory likely does not depend on the activity of time cells, especially for delays greater than a few seconds. Overall, the question of whether hippocampal (time cell) activity during delay periods is relevant for working memory is interesting. However, answering this question would require more directed, experimental approaches (e.g., directly manipulating neurons active during the delay). Instead, the results presented here are more correlational and leave many questions open when relating time cell activity to working memory. Finally, several recent studies that have contributed to refining our understanding of the relationship between time cells, theta, and memory still need to be included here.

Major comments:

- In their abstract, the authors state that they '[...] examine whether time cells during WM maintenance depend on ongoing theta oscillations [...]'. To this end, they compare a condition where animals run on a treadmill (theta state) and a treadmill-off condition (less theta). Previous studies have employed pharmacological approaches (Wang et al., 2015; cited), thermal cooling (Petersen & Buzsaki, 2020; not cited), or optogenetics (Eter et al., 2023; not cited) to control for theta oscillations in a similar task. The Petersen paper, in particular, proposes a phase-based explanation for how disrupting theta could lead to altered memory retrieval, which could be critical in discussing the results presented by the authors. While correlational approaches also have their value, we are still determining how this paradigm could tell us something novel about the relationship between theta and time cells (especially given that the treadmill-off condition is not entirely devoid of theta oscillations).

- The authors often omit several key findings that are directly relevant to their study or cite studies but do not contextualize them with their own study. For instance, L33: the observation that time cells are disrupted by pharmacological manipulation of the medial septum by Wang et al. (2015; cited) has to be balanced with more recent studies showing that lesion-based (as well optogenetic) disruptions of theta do not alter time cell activity (they do cite the Sabariego et al., 2019 paper, but not in this context). Moreover, recent experiments silencing septohippocampal GABA cells, which drastically reduced theta, produced a relatively small change in time cell representations during the delay (in a T-maze) but had no effect on working memory performance (Yong et al., 2022; bioRxiv 2022.06.25.497592). This suggests that significant reductions of theta power in the delay have relatively little effect on both time cells and working memory, further weakening the possible role of time cells in working memory. Overall, this is also where we fail to establish the advance of this paper: the current knowledge suggests that sequential activity in CA1 is dissociated from working memory function in several studies. While it is interesting that the authors investigate a condition where animals do not run (treadmill-off), we are unclear on whether they hypothesized that such activity could be differentially related to memory function

- In Fig. 1D, we need clarification about why neurons are sorted by their peak activity location (instead of time since this is what is then analyzed in E). While rats run on the treadmill, should not one area of the 'delay zone' be over-represented (since the animal is running in place)?

- The authors mention that the delay zone is over-represented, but I could not find any direct quantification for that (with statistics)
- The authors often provide examples but no detailed quantification. For example, in Fig. 3, the authors describe two types of firing patterns and provide some excellent examples. However, could they devise a metric that would allow them to discern those two firing patterns in an unbiased (instead of manual) manner? Then, given the two unbiased groups, provide statistics for when these are preferentially expressed.
- The authors state that cell activity during the delay neither informs past nor future decisions. First, we are surprised as sharp-wave ripples were previously shown to contain information about previous spatial trajectories. These are likely to occur in the treadmill-off condition. Given their statement, could the authors show an example of ripple activity in the treadmill-off condition and test whether this activity is related to previous explorations?
- Related to this previous comment, we wonder whether the authors can really conclude about the lack of 'informativeness' of time cells given their current analyses. In Fig. 5, we were expecting a quantification of replayed trajectories by time cells representing previously explored locations (see Wang et al., 2015), which seems to be missing.

Minor comments

- Line 23: the authors tend to conflate results found in the prefrontal cortex to those in the hippocampus. Since the authors focus on hippocampal activity, those prefrontal studies might not be of direct relevance in their argumentation

(Remarks on code availability)

Reviewer #2

(Remarks to the Author)

In this manuscript, the authors try to address an important long-standing question in the field of hippocampal research: What is the relationship between theta oscillations and temporally-tuned firing fields (time fields)? Previous evidence suggests that theta oscillations are necessary for task performance in hippocampus-dependent memory tasks such as the delayed alternation task, and for the occurrence of time cells.

To investigate this, the authors compared trials in which animals ran on a treadmill with trials where the treadmill was turned off during the delay period. They observed that, although the theta amplitude was low in the treadmill-off trials, time cells were still present. Additionally, they found that a limited number of cells exhibited differential firing activity in the left-turn vs. right-turn trials, suggesting that these cells cannot predict the trial types.

The results are interesting. There are questions regarding the comparison between the current study and previous findings.

For the trials where the treadmill is on, it is unclear whether the animal had to run for 10 or 30 seconds continuously or they were allowed to stop intermittently. Based on the theta bout durations shown in Figure 1C, it seems that the animals can stop during the run. If so, do the animals need to accumulate 10 or 30 seconds of running time, or does the time counter continue even when the animal is not running?

In the Pastalkova et al., 2008 paper, animals are required to run continuously and steadily over the 10-20 second delay. This difference may explain the difference in the time cell abundance after 5-second delay.

Another related question: In the treadmill-on condition, the theta amplitude drops after ~5 seconds. How much of this drop is related to running speed change over the delay period? What is the running speed profile over the delay period? In the treadmill-off block, the theta amplitude drops after 2-3 seconds on average. Can the firing fields preferentially appear in trials where the initial theta bout is longer?

In the MacDonald et al., 2014 time cell paper, time cells are present beyond the 5-second delay, even though the animal is not moving much in the delay area. In that study, theta oscillations existed throughout the delay period. This finding, together with the results from the Pastalkova 2008 paper, implies that potentially continuous theta might play a role in producing continuous sequences across the delay period.

In addition, both papers mentioned above show clear differential firing between either left- and right-turn trials or trials with different object and odor pairs. In the Pastalkova et al. paper, the animals were first trained for 3-second wheel running before starting the alternation task training. In this manuscript and the Ito et al., 2015 paper, it seems that the animals were trained for alternation before introducing the delay. It is unclear whether the difference in training protocols might lead to differences in the occurrence of turn-selective firing cells.

There also seems to be a large difference in terms of the percentage of neurons with fields in the delay area, between the

current study (~ 10%) and the Pastalkova (32%), or MacDonald (53%) papers. The percentage of time cells is even lower in the animals trained specifically for the treadmill on task. What might be the reasons for this discrepancy? Do the time cells recorded in the current study show clear phase precession?

There appear to be more cells persistently active in the 10-second delay trials compared to the 30-second delay trials. If persistently active cells are simply an extension from the time cells, shouldn't we expect to see a similar proportion of cells in the 10-second and 30-second conditions?

If I understand this correctly, in Figure 4A, the bottom row contains all the persistently active cells. It looks like these cells can start firing at any time during the delay, not necessarily after 5 seconds. Also, if the median peak time is 4.7 seconds, it is not very convincing to claim that persistently active cells mainly appear after 5 seconds.

From S6, it looks like there are still some theta oscillations in the treadmill off condition.

Figures 5E and 5F are in agreement with the report in the Ito et.al. 2015 paper on that more turn-selective cells occur in the stem area than in the delay area.

(Remarks on code availability)

Reviewer #3

(Remarks to the Author)

This manuscript examined the activity of hippocampal cells in delay periods in between spatial working memory trials on a continuous T-maze. Two alternative delay periods of 10 or 30 seconds were used, and animals either ran on a treadmill or remained stationary. They were able to differentiate two different types of cells: those that fired within 5sec with a relatively fixed interval from the delay start and another that exhibited persistent firing during the delay (delay active cells). Cell populations that were active in treadmill running and non-running conditions were different. Surprisingly, the activity of the cells during the delay did not predict the future arm choice of the animal, suggesting that the hippocampus was not involved in 'rehearsing' short-term spatial memory in this task during the delay. The work resolved multiple unresolved issues related to 'time cells', including the dependence of theta oscillations and delay length. Importantly, it demonstrated that delay time-dependent activity could be observed during relatively shorter delay of <5 s.

Overall, I have a few technical questions:

Major comments

Was the lateral position of the animal controlled for when it was examined whether, at the choice point, the activity of cells predicted which arm the animal chose?

I assume interneurons were not included in the delay-active cells. At least in Fig2e, some of these had ~10Hz rates. Also, it would be useful to show that the firing rate of these cells drops during maze running.

Was it quantified whether the rates of the 'time cells' were similar at the 10 vs. 30 s delay conditions? Were the time intervals also similar when they were active during the delay? Perhaps it was quantified, but I cannot find an analysis related to this on the figures.

Would it be possible to check whether activity in the delay periods during error trials was similar to that of correct trials? This could provide additional evidence that hippocampal activity during the memory delay is not involved.

This may have been mentioned in the results, but did this study observe spitter cells in the central arm? Some work (e.g., Emma Wood's) did not observe spitter cells during delayed alternation tasks, while others have seen this. The late Howard Eichebaum suggested that depending on how animals are trained, animals may follow predetermined paths from the delayed periods on, or they may decide only at the decision point which way they go next. The strategy of the animal could be inferred based on the presence of spitter cells or testing whether the animals consistently stopped at the choice point.

Minor

Although the results are clearly written, it is quite boring to read, partially because many of the less important results are discussed in the main text without figures. Perhaps making supplementary figures about less relevant results and only briefly referring to them in the main text would make the text more concise and convey the underlying story better.

(Remarks on code availability)

Reviewer #4

(Remarks to the Author)

In this study, Yuan and colleagues set out to study hippocampal activity during the delay interval of a working memory task

and its dependence on theta oscillations. They use a behavioral design which is very well suited to address their question: a spatial alternation task with interleaved blocks of "treadmill on" and "treadmill off" conditions, as well as 10- vs 30-second delay durations. They show that sequential firing of neurons in CA1 during a delay period before choice is not influenced by the amount and power of theta oscillations. They also convincingly show that sequential firing takes place in the first few seconds of the delay, while cells active afterwards tend to remain active for the rest of the delay period. They then provide evidence suggesting the absence of trajectory-dependent coding during the delay interval both when mice are required to run and when they are not. Previous work from the same group (Sabariego, Marta et al. 2019) had previously showed that time cells active during the 10-s delay period of the same alternation task did not distinguish between left-turn and right-turn trials; the authors replicate this finding in the manuscript for both type of trial conditions and for assembly activity in addition to single cells activity, making this a strong observation. Finally, the authors analyze large population events happening during the delay interval; they propose that activity during these events did not code for turn direction, and is therefore uninformative about past behavior or future choice. However, the evidence shown in support of this conclusion appears for now a bit preliminary; their data shows that sequences detected during the delay are more similar among each other when compared to sequences detected during reward population events. More evidence is needed to justify this conclusion. Overall, this work will be of general interest to the field of. Since the work challenges in part what was found in previous literature, it would be useful for readers to have a more exhaustive discussion of possible reasons for the differences and discrepancies with previous findings.

Specific comments:

-In figure 1B, what is the reason for the use of theta-delta ratio when quantifying the difference in theta power between conditions? Could the authors include theta spectrograms?

- In figure 1D and 3D: why is the firing distribution across the maze (and relative quantifications) shown for all animals together? It is more appropriate to show them separately per animal.

-Could the authors speculate on how to explain the difference with previous studies? One point to be addressed is whether animals might be overtrained (3-5 weeks of training with continuous alternation + 10 days with delay), and if that could influence the results.

-Given that the authors themselves speculate, earlier in the manuscript, that interleaving trials with and without running could make a difference, compared to just training animals with one trial type, the analysis shown in fig.5 should be repeated for animal groups with running trials only, in order to make the claim more solid.

- In order for results in fig.6 to support the author's claim that "Large-scale population events in the delay zone did not code for turn direction", only comparing delay and reward events as a whole does not seem sufficient. For instance, the same analysis of temporal activation patterns could be performed, but using activity in half of the trials before turns in one direction to create their template, and use that as a comparison with the other half, as well as to turns in the other direction.

Minor comments:

-The difference in the percentage of turn-selective cells and assemblies between the two conditions (treadmill on and treadmill off) for activity during the stem (fig 5F) seems puzzling, given that the behavioral state of the animals should be comparable at that point in time How could that be explained?

- Regarding the analysis of population events (fig.6), could the authors justify the reasons why population events were defined as they did, with event onset defined as the first spike time of any neuron active during the event itself, rather than by detecting ripples from the LFP?

Moreover, data regarding the number of populations events during reward and delay epochs (both as a whole and per animal) is missing.

-It would be useful to allow readers to compare, throughout the manuscript, data coming from individual animals, to judge their consistency and variability.

(Remarks on code availability)

Reviewer #5

(Remarks to the Author)

I co-reviewed this manuscript with one of the reviewers who provided the listed reports. This is part of the Nature Communications initiative to facilitate training in peer review and to provide appropriate recognition for Early Career Researchers who co-review manuscripts."

(Remarks on code availability)

Version 1:

Reviewer comments:

Reviewer #1

(Remarks to the Author)

The authors have clearly addressed this reviewer's questions and provided a substantial amount of data, including additional experiments, to support the queries. The manuscript makes an additional contribution by dissociating the role of time cells in a working memory task.

(Remarks on code availability)

Reviewer #2

(Remarks to the Author)

The authors have done a great job and significantly improved the manuscript.

One point that may benefit from further discussion is the relationship among running speed, theta oscillations, time cells, and working memory. A substantial body of evidence indicates that both the power and frequency of hippocampal theta oscillations scale positively with running speed. Prior studies that robustly reported time cells, such as Pastalkova et al. (Science, 2008) and MacDonald et al. (Neuron, 2013), involved relatively high running speeds—exceeding 50 cm/s during wheel running and ranging from 35–49 cm/s on a treadmill, respectively. In both cases, time cells were observed across extended delay periods of up to 15 seconds.

By contrast, in both the current study and the earlier work by Yong et al. (bioRxiv, 2022), animals maintained a much lower running speed of ~20 cm/s. Notably, these studies concluded that time cells are not essential for successful performance in working memory tasks. The discrepancy in running speed may provide a plausible explanation: slower movement could result in weaker theta engagement, thereby limiting the recruitment or temporal span of time cells, especially beyond 5 seconds. It is therefore plausible that the reduced speed in the present task both lowers cognitive demand and constrains the contribution of time cells to task performance.

Finally, while the observed increase in the proportion of time cells with less training is an interesting result, differences in experience may not fully explain the variability in time cell prevalence. For instance, animals in Pastalkova et al. (2008) also reached stable performance within approximately one week—comparable to the timeline in the current study.

(Remarks on code availability)

Reviewer #3

(Remarks to the Author)

The revision has addressed the issues I have raised and my questions.

(Remarks on code availability)

Reviewer #4

(Remarks to the Author)

The authors have answered all my comments. This work will have an important contribution to the field.

(Remarks on code availability)

Reviewer #5

(Remarks to the Author)

(Remarks on code availability)

Point-by-point responses, Yuan et al., NCOMMS-24-30026

We sincerely thank the editor and five reviewers for the time and expertise dedicated to evaluating our manuscript. We highly appreciate the detailed sets of comments, and the opportunity to address the reviewer suggestions to clarify and expand on many points that were not satisfactory in the original version. We have thoroughly addressed the concerns by carrying out additional experiments, by making extensive revisions, and by adding numerous figure panels.

Key additions/revisions are highlighted here and detailed point-by-point responses are provided in the remainder of the document:

-We expanded/replaced many of the main figures (e.g., panels b-f, i, j in Fig. 1, all panels in Fig. 2, panel d in Fig. 4, panel a in Fig. 5, all panels in Fig. 6 and panels a, b, d-g in Fig. 7) and increased the number of supplementary figures from 8 to 14.

-The revisions approximately double the extent of previously already comprehensive data analyses. For example, new measurements of theta and behavior during the delay interval are added, as well as key analyses for population events and their association with sharp-wave ripples.

-New Neuropixels recording data from 6 additional rats are included, which show that the fraction of time cells is indeed dependent on the amount of training. Despite the wide range of time cell fractions across training extent, we nonetheless observe qualitatively similar results across a large number of rats (a total of 19 in the revised manuscript) and continue to show that sequential firing predominantly occurs during the first few seconds of the delay.

The additional experiments and analyses further strengthen the key conclusions of the manuscript, which is that hippocampal sequential firing patterns are short-lasting can thus be not a common mechanism for working memory retention over longer retention intervals. We reasoned that the extensive revisions in response to reviewer comments warrant a title change from 'Time cell sequences during delay intervals are not dependent on brain state' to 'Time cell sequences during delay intervals are not dependent on brain state and do not support hippocampus-dependent working memory'.

Again, we thank the reviewers for their thoughtful comments, which have prompted new recordings and many analyses that led to a much-improved manuscript on the extent and limitations of sequential firing patterns in the hippocampus. We hope that the thoroughly revised and improved manuscript will now be accepted for publication in *Nature Communications*.

Point-by-point response to reviewers are included below, with reviewer comments retained in black and our responses added in blue. A revised manuscript with key revisions highlighted in red is included with the resubmission.

Reviewer #1 (Remarks to the Author):

The authors investigate whether time cells can be active outside of periods of theta activity. To this end, they propose a task where rats are forced to run or rest (run periods associated with theta oscillations). They found that time cells mainly represented the first few seconds of each delay period and that these neurons were not related to working memory. Populations of time cells were also state-specific and less affected by the duration of the delay. They conclude that working memory likely does not depend on the activity of time cells, especially for delays greater than a few seconds. Overall, the question of whether hippocampal (time cell) activity during delay periods is relevant for working memory is interesting. However, answering this question would require more directed, experimental approaches (e.g., directly manipulating neurons active during the delay). Instead, the

results presented here are more correlational and leave many questions open when relating time cell activity to working memory. Finally, several recent studies that have contributed to refining our understanding of the relationship between time cells, theta, and memory still need to be included here.

Major comments:

- In their abstract, the authors state that they '[...] examine whether time cells during WM maintenance depend on ongoing theta oscillations [...]'. To this end, they compare a condition where animals run on a treadmill (theta state) and a treadmill-off condition (less theta). Previous studies have employed pharmacological approaches (Wang et al., 2015; cited), thermal cooling (Petersen & Buzsaki, 2020; not cited), or optogenetics (Etter et al., 2023; not cited) to control for theta oscillations in a similar task. The Petersen paper, in particular, proposes a phase-based explanation for how disrupting theta could lead to altered memory retrieval, which could be critical in discussing the results presented by the authors. While correlational approaches also have their value, we are still determining how this paradigm could tell us something novel about the relationship between theta and time cells (especially given that the treadmill-off condition is not entirely devoid of theta oscillations).

We may not have sufficiently emphasized that the treadmill-off condition is certainly not devoid of theta. The key observation for the treadmill-off condition is the discontinuity of theta oscillations, rather than the modest decrease in theta power. Because of the discontinuity, we hypothesized that sequential neuronal activity patterns terminate at approximately the time of the first major pause in theta in the treadmill-off condition, and conversely, extend throughout the delay period during uninterrupted theta oscillations in the treadmill-on condition. This hypothesis is based on the previously reported contradictory findings that we observed short lasting firing sequences (< 5 s) without running during the delay (Sabariego et al., 2019) while others observed prolonged sequential firing patterns (> 5 s) with running during the delay. Given our previous result of unremarkable/short-lasting time cell sequences without running and the replication of these results in the present study (Figs. 4d and 5b), we did not consider a further decrease in theta oscillation amplitude or duration an informative approach for examining the question whether the duration of sequential firing can be prolonged by sustained theta oscillations. Rather, we reasoned that we ought to elicit uninterrupted theta oscillations to observe longer lasting sequences. If our hypothesis were supported, we would obtain evidence that longer theta bouts would lead to long-lasting sequential firing of hippocampal cell populations.

Fig. 4d:

Fig. 5b:

We are of course well aware that hippocampal sequential firing/neuronal activity can be diminished by pharmacological approaches. However, given that we only observed short-lasting time cell sequences without running during the delay (Sabariego et al., 2019), our goal was to not further curtail time cell sequences, but to prolong them to test our hypothesis. Neither pharmacological, cooling or inhibitory optogenetic approaches would thus be suitable, given that they result in reducing/slowing theta oscillations. We could in theory have used optogenetic methods to pace inhibitory cells in the medial septal area (MSA) to generate long-lasting oscillations over periods of seconds and minutes, in particular since we and others successfully used the method (Zutshi et al., 2018; Quirk et al., 2021; Lepperød et al., 2021; Etter et al. 2023). However, the artificial pacing of MSA inhibitory neurons results in substantial hypersynchronization of the entorhinal cortex and hippocampus (Quirk et al., 2021; Lepperød et al., 2021; Etter et al., 2023). Given that hypersynchronization would result in synchronous activity patterns of hippocampal cell populations within the theta cycle, while we would want to preserve sequential activity patterns within each theta cycle, there was no compelling rationale for deploying approaches that are known to diminish the precise temporal order of neuronal activity patterns. To our knowledge, **sustained running is the only currently available manipulation that elicits uninterrupted theta oscillations without concurrently mistiming the precise temporal firing patterns in the entorhinal-hippocampal circuit.** We therefore consider running on a treadmill the method of choice and are not aware of any other manipulation that could result in comparable increases in the length of theta bouts without undesirable collateral effects.

We fully agree and acknowledge that correlational approaches have limitations, but would want to point out that our approach is not purely correlational, but **includes a behavioral manipulation that leads to major changes in LFP patterns without concerns about off-target effects.** Irrespective of whether our manipulation is considered causative or correlational, we do not find that sustained theta oscillations result in/are correlated with prolonged sequential activity. Even if this is interpreted as a correlational result, the findings conclusively show that there is no relation between how long theta oscillations and how long time cell sequences last. Further reducing theta oscillation amplitude or frequency, in particular with methods that have other unintended effects (e.g., reduced firing rates), would not more comprehensively make the point that there is no relationship between the duration of theta epochs and the length of time cell sequences.

Petersen et al. provide interesting data on the coupling of cells to theta oscillations. When theta frequency was lowered by septal cooling, phase coupling of spiking was retained, even though this necessarily resulted in larger time differences (i.e., on a millisecond time scale) between cells that were active within the same theta cycle. Furthermore, our reading of their results is that fixed phase/stretched time did not have effects on sequential firing patterns within the theta cycle. Because there was no delay period in their task, they could not test effects of the manipulation on time cells. Therefore, the result by Petersen et al. show preserved within-cycle theta sequences along with a memory impairment, while our results demonstrate the absence of time cell sequences during the delay interval (with or without sustained theta) along with intact memory. As such, we find it challenging to directly relate these findings except to conclude that there is no correspondence between sequential firing patterns either within the theta cycle (their data) or on a longer scale (our data) and memory performance. We now further emphasize this conclusion by describing and citing the results of Petersen et al.

Line 366-369:

'Along with the finding that within-cycle theta sequences can be preserved when spatial memory is impaired³⁰, there is thus no consistent correspondence between sequential firing patterns—either within the theta cycle or within the delay interval—and memory performance.'

If the reviewer had a different conceptual framework in mind for how the results by Petersen et al. could relate to ours, we apologize that we are not able to develop the same line of reasoning and would like to hear further details on how we could relate the results by Petersen et al. to ours.

- The authors often omit several key findings that are directly relevant to their study or cite studies but do not

contextualize them with their own study. For instance, L33: the observation that time cells are disrupted by pharmacological manipulation of the medial septum by Wang et al. (2015; cited) has to be balanced with more recent studies showing that lesion-based (as well optogenetic) disruptions of theta do not alter time cell activity (they do cite the Sabariego et al., 2019 paper, but not in this context). Moreover, recent experiments silencing septohippocampal GABA cells, which drastically reduced theta, produced a relatively small change in time cell representations during the delay (in a T-maze) but had no effect on working memory performance (Yong et al., 2022; bioRxiv 2022.06.25.497592). This suggests that significant reductions of theta power in the delay have relatively little effect on both time cells and working memory, further weakening the possible role of time cells in working memory. Overall, this is also where we fail to establish the advance of this paper: the current knowledge suggests that sequential activity in CA1 is dissociated from working memory function in several studies. While it is interesting that the authors investigate a condition where animals do not run (treadmill-off), we are unclear on whether they hypothesized that such activity could be differentially related to memory function

We concur that the mentioned paragraph previously included only some of the conflicting results and not the full list of citations that is provided above. We therefore substantially rewrote the paragraph to include the studies that failed to find effects of manipulation of theta oscillations on sequential firing patterns. However, we suggest that our previous study (Sabariego et al., 2019) needs to be carefully interpreted in this context. There were already few time cells in controls, which makes it challenging to ascertain whether the entorhinal lesion resulted in a further reduction.

Line 35-45:

'Despite the possible relation between theta oscillations and sequential firing over time scales of many seconds, reports on the effects of reducing theta oscillations on time cells are inconsistent. Hippocampal time cells are diminished when theta amplitude is substantially reduced by pharmacological silencing of the medial septal area (MSA)¹⁸, but retained when theta amplitude is reduced with optogenetic silencing of MSA¹⁹. Similarly, in spatial alternation tasks with no requirement for animals to run during the delay interval and thus with reduced theta oscillations, hippocampal time cell sequences last only for a few seconds¹⁶. Evidence on the persistence of time cells at low levels of theta oscillations is therefore inconsistent, and conversely, these studies have not established whether increasing or prolonging theta oscillations is necessary for extending the time period over which sequential activity is sustained during WM.'

Here, we now also mention that the results by Yong et al. (2022), who reported preserved time cell sequences, are conflicting with those by Wang et al. (2015), who reported that time cell sequences are disrupted by lowering theta oscillations. These differences can perhaps be reconciled by considering differences in the methods, where selective silencing of septal GABAergic cells (Yong et al., 2022) may have more limited effects than the non-selective silencing of the medial septal area by muscimol (Wang et al., 2015), even though both manipulations decrease theta oscillation amplitude.

Although we would like to cautiously interpret studies that are not yet peer-reviewed (Yong et al., 2022), we respectfully disagree with the suggestion that the only plausible interpretation of their results is that sequential activity is dissociated from working memory function. To the contrary, **the finding that time cells and working memory are retained in conditions with reduced theta could be interpreted as implying that sequential activity on a behavioral scale is critical for working memory performance.** In particular, the results are consistent with the interpretation that it is not theta oscillations that are relevant for working memory, but rather sequential activity. Accordingly, their result would further support the widely held conjecture that time/sequence cells are critical for working memory retention over the delay interval, as originally reported by Pastalkova et al. (2008).

With all of these studies (Pastalkova, 2008; Wang, 2015; Yong, 2022) pointing towards the possibility that sequential activity of hippocampal cells on a behavioral time scale may be critical for working memory performance, we see the statement that 'sequential activity in CA1 is dissociated from working memory function

in several studies' only supported for theta sequences (i.e., within a theta cycle; Petersen, 2020), but not for sequences on the behavioral time scale. **Our study therefore provides an important advance compared to the previous studies, which is the dissociation between sequential activity on a behavioral scale ('time cells') and memory performance.** This is now further emphasized in our discussion.

Line 361-369:

'Sequential activity patterns thus have a limited lifetime and are not sustained across long delay intervals, even in conditions when theta oscillations are ongoing throughout the delay. Because behavioral performance was high with both the 10-s and the 30-s delay, these data already suggest that sequential firing patterns throughout the delay are not necessary for memory retention, including in conditions with uninterrupted theta oscillations. Therefore, there were no sustained time cell sequences during the delay interval—with or without sustained theta—when memory was intact. Along with the finding that within-cycle theta sequences can be preserved when spatial memory is impaired³⁰, there is thus no consistent correspondence between sequential firing patterns—either within the theta cycle or within the delay interval—and memory performance.'

- In Fig. 1D, we need clarification about why neurons are sorted by their peak activity location (instead of time since this is what is then analyzed in E). While rats run on the treadmill, should not one area of the 'delay zone' be over-represented (since the animal is running in place)?

This panel was meant to show that delay-active hippocampal cells, when active in zones other than the delay zone, have standard spatial firing patterns throughout the maze. This is the reason why we sorted by the spatial peak activity location. However, displaying the data as originally shown, with the sorting including the delay, did not rigorously reflect the order of spatial firing fields outside of the delay zone.

Also note that activity in the delay zone does not necessarily translate into an obvious spatial peak at the most densely occupied site in the delay zone. By definition, time cells can be active briefly (e.g., 1-2 s), which will result in a relatively low peak rate at a spatial bin when dividing the number of spikes by the time spent at the location. The spatial plots are therefore not informative about a cell's firing location in the delay zone.

To more thoroughly characterize the spatial and temporal firing patterns of delay-active cells, we have now substantially modified and expanded the panels that display the spatial and temporal firing patterns of delay-active cells. First, new figure panels and a supplementary figure are now included that precisely show the occupancy within the delay zone (for an example rat: Fig. 1c; for all rats: Fig. S1). As expected, the majority of the time in trials with the treadmill on was spent with the head in a fixed position in front of the barrier. When rats had the option to freely move in the delay zone during treadmill-off trials, head position was more distributed. Next, we now show the spatial distribution of firing peaks separately for cells that are not active in the delay and for delay-active cells (Fig. 2c). These plots reveal that spatial firing outside of the delay zone covers the entire maze and that this is the case for cells that are not active in delay and for delay-active cells. In addition, the area close to the barrier can be overrepresented by delay-active cells, as predicted by the reviewer, although this is not consistent for the reason stated in the previous paragraph.

Fig. 1c:

Fig. 2c:

Fig. S1:

Figure S1

- The authors mention that the delay zone is over-represented, but I could not find any direct quantification for that (with statistics)

This information was not prominently shown (only in the previous Fig. S1d). We agree that this is an important point and therefore moved the key panels from the supplementary figure to a main figure (now Fig. 2a and b). In Fig. 2b, it can be seen that the size of the remainder of the maze is ~4 fold larger than the size of the delay zone and that, in 10-s delay trials, occupancy in the remainder of the maze is ~3 fold higher than occupancy in the delay zone. Yet, the proportion of active cells in the remainder of the maze compared to the delay zone is only ~1.5 fold higher (Fig. 2a).

Fig. 2a and b:

In addition to moving the figure panels to the main figure, we also rewrote the corresponding results section to better make the point that there are major discrepancies between relative size, relative occupancy, and proportion of active cells.

Line 97-112:

'In each treadmill/delay condition, the proportion of putative pyramidal cells that were active (average rate > 0.5 Hz) in maze areas other than the delay zone was ~1.5 fold higher than the proportion active in the delay zone (% cells active in remainder of the maze, on/10-s: 46.5% ± 5.6; off/10-s: 42.6% ± 5.3; on/30-s: 42.5% ± 6.3; off/30-s: 43.9% ± 5.6; % cells active in delay, on/10-s: 32.3% ± 3.2; off/10-s: 34.6% ± 4.2; on/30-s: 31.2% ± 4.1; off/30-s: 34.6% ± 4.9; mean ± SEM, n = 18 sessions from 5 rats; Figure 2a). In comparison, the size of the remainder of the maze was ~4-fold larger than the delay zone (Figure 2b). Therefore, the proportion of cells active in the delay area was higher than predicted from the relative size of the delay zone (delay zone, ratio of % active cells by % size of zone, on/10-s: 1.6 ± 0.16; off/10-s: 1.7 ± 0.21; on/30-s: 1.6 ± 0.20; off/30-s: 1.7 ± 0.24, mean ± SEM; remainder of the maze: on/10-s: 0.58 ± 0.07; off/10-s: 0.53 ± 0.07; on/30-s: 0.53 ± 0.08; off/30-s: 0.55 ± 0.07, mean ± SEM; F(1,143) = 112.48, p = 1.6x10⁻¹⁹, n = 18 sessions from 5 rats, ANOVA: two-factor with replication), similar to the pattern that has been described in the running wheel⁸. The overrepresentation of the delay area could be explained by the amount of time in the delay zone rather than the spatial size of the delay zone. However, irrespective of the proportion of time that was spent in the delay zone (10-s delay: 26.6% ± 0.92; 30-s delay: 50.0% ± 1.0; mean ± SEM; n = 18 sessions; Figure 2b), the proportion of delay-active cells remained approximately the same (F(3,68) = 0.18, p = 0.91, ANOVA; Figure 2a).'

- The authors often provide examples but no detailed quantification. For example, in Fig. 3, the authors describe two types of firing patterns and provide some excellent examples. However, could they devise a metric that would allow them to discern those two firing patterns in an unbiased (instead of manual) manner? Then, given the two unbiased groups, provide statistics for when these are preferentially expressed.

We did distinguish time-limited cells and persistently active cells by requiring that the former be active at < 20% of their peak firing rate in any bin after their peak and that the latter be active at ≥ 20% of their peak firing rate in all bins throughout the remainder of the delay. The criterion for classifying the cells of both types was

previously mentioned in the methods section, but in the main text only for the time-limited cells. The criterion for the persistently active cells is now added in the results section.

Line 168-171:

'Because these results are consistent with two distinct temporal firing profiles, we further divided time cells into time-limited cells (firing rate < 20% of peak rate in any time bin between the peak and the end of the delay) and into persistently active cells (firing rate ≥ 20% of peak rate in all time bins between the peak and the end of the delay).'

We also previously reported statistics to compare whether the proportion of the two cell types changed between the treadmill-on and treadmill-off conditions.

Now line 174-179:

'We identified a similar fraction of time-limited cells in all delay/treadmill conditions (on/10-s: 11.18% ± 2.19; off/10-s: 9.09% ± 2.4; on/30-s: 10.74% ± 1.89; off/30-s: 12.24% ± 1.92, mean ± SEM, n = 18 sessions; F(3,68) = 0.41, p = 0.75, ANOVA; Figure S6a and b, Table S2 and S3), and most time-limited cells had finely tuned time fields with peak firing during the initial 5 s of the delay interval (on/10-s: 98.46% ± 1.6; off/10-s: 100.0% ± 0.0; on/30-s: 84.88% ± 7.92; off/30-s: 82.75% ± 8.54, mean ± SEM, n = 18 sessions; Figure 5a-c, Figure S6c and d).'

Now line 236-239:

'Similar to what we observed for time-limited cells, the fraction of persistently active cells did also not differ between treadmill-on and off conditions (on/10-s: 8.87% ± 1.78, off/10-s: 9.78% ± 2.50, p = 0.74, paired t-tests, n = 18 sessions; on/30-s: 5.13% ± 1.28, off/30-s: 3.72% ± 0.93, p = 0.39, paired t-tests, n = 18 sessions; Figure 6a-d).'

If the statistics that we are providing are not corresponding to the requested comparisons of 'when these are preferentially expressed', we would appreciate further guidance on which conditions to compare.

- The authors state that cell activity during the delay neither informs past nor future decisions. First, we are surprised as sharp-wave ripples were previously shown to contain information about previous spatial trajectories. These are likely to occur in the treadmill-off condition. Given their statement, could the authors show an example of ripple activity in the treadmill-off condition and test whether this activity is related to previous explorations?

Given that theta oscillations were discontinuous, but not completely absent in the treadmill-off condition, the proportion of ripples during the delay interval was relatively low. We nonetheless observed that brief population events (i.e., high fraction of active cells) were often accompanied by sharp-wave ripples (see examples in Fig. 7c and e and Fig. S11d). To further expand on this analysis, we now measured the extent to which SWRs and population events overlapped (Fig. 7d and e). A comparison of LFP (i.e., SWRs; 3 SD above the mean power in 150-250 Hz band) and neuronal firing patterns (i.e., population events) revealed that most population events were associated with sharp-wave ripples (54.5%/52.6% in delay/reward). Conversely, the proportion of ripples that were accompanied by population events was somewhat lower (29.5%/48.8% in delay/reward), although this will of course depend on the number of simultaneously recorded cells. Because we could only meaningfully analyze temporal order when a large fraction of cells was co-active, we retained the analysis of firing patterns in population events and now point out in the text and figure legend that most of these were accompanied by SWRs.

Fig. S11d:

Fig. 7:

Line 276-278:

'We confirmed that most population events in the delay interval and at the reward locations were accompanied by sharp-wave ripples (SWRs; 54.5%/52.6% in delay/reward; Figure 7d and e).'

As stated, we did previously not include standard analysis to determine whether cells were reactivated or trajectories are replayed. To test whether the activity in population events was related to the immediately preceding trajectory on either the right or the left side of the maze, we tested whether cells with place fields on either right side or left side were, following trajectories from either the right or left side, selectively reactivated during population events in the delay zone. This analysis was done in lieu of analyzing replayed trajectories more directly, which was not feasible with the cell numbers in our study. As described in more detail in response to the next point, reactivation analysis of cells during population events is now added (Fig. 7f), and compared to selective activation during the entire delay period (Fig. 7a).

- Related to this previous comment, we wonder whether the authors can really conclude about the lack of 'informativeness' of time cells given their current analyses. In Fig. 5, we were expecting a quantification of replayed trajectories by time cells representing previously explored locations (see Wang et al., 2015), which seems to be missing.

We hope to correctly interpret this comment and assume that it refers to the reactivation of cells in the delay zone (e.g., time cells) that correspond to place cell sequences in either the right or left side of the maze. If place fields were traversed as part of a trajectory along a maze arm, the same cells would be reactivated during replay of the trajectory. Although we observed a substantial number of place cells on either the left or right arm, we did not observe that these were then again selectively active in population events during the delay in either left-turn or right-turn trials (Fig. 7f). This result was also observed more broadly when analyzing whether assemblies were differentially active during the delay interval (Fig. 7a), but note that cells could be part of an assembly irrespective of whether they had spatially selective activity elsewhere on the maze. In contrast, population events in the reward area, which are often associated with SWRs (Fig. 7d), had a high proportion of cells that were selectively reactivated at only one or the other reward location (Fig. 7f).

These results are now shown in Fig. 7f and described in the text.

Line 278-282:

'However, we did not find turn-selective reactivation above chance for population events in the delay zone (off/10-s: $p = 0.090$, $n = 92$; off/30-s: $p = 0.064$, $n = 194$). As a control, we analyzed left vs. right selective reactivation at the reward locations and detected clear differences (on/10-s: $p < 0.001$, $n = 86$; off/10-s: $p < 0.001$, $n = 139$; on/30-s: $p < 0.001$, $n = 92$; off/30-s: $p < 0.001$, $n = 86$; Figure 7f).'

Minor comments

- Line 23: the authors tend to conflate results found in the prefrontal cortex to those in the hippocampus. Since the authors focus on hippocampal activity, those prefrontal studies might not be of direct relevance in their argumentation

The sentence is now rewritten to no longer refer to the prefrontal studies, and the citations were removed.

Line 20-26:

'Sequential hippocampal activity patterns during delay intervals have been shown to correlate with successful behavior outcomes in WM tasks that depend on hippocampus and entorhinal cortex^{8, 13-16}. In particular, studies that reduced hippocampal theta oscillations—the predominant local field potential (LFP) during running¹⁷—demonstrated that the extent of remaining sequential firing was closely related to memory performance^{18, 19}. However, similar subsequent studies also assessing population activity in WM tasks could not detect a relationship between hippocampal delay activity and spatial WM performance^{16, 20, 21}.'

Reviewer #2 (Remarks to the Author):

In this manuscript, the authors try to address an important long-standing question in the field of hippocampal research: What is the relationship between theta oscillations and temporally-tuned firing fields (time fields)? Previous evidence suggests that theta oscillations are necessary for task performance in hippocampus-dependent memory tasks such as the delayed alternation task, and for the occurrence of time cells.

To investigate this, the authors compared trials in which animals ran on a treadmill with trials where the treadmill was turned off during the delay period. They observed that, although the theta amplitude was low in the treadmill-off trials, time cells were still present. Additionally, they found that a limited number of cells exhibited differential firing activity in the left-turn vs. right-turn trials, suggesting that these cells cannot predict the trial types.

The results are interesting. There are questions regarding the comparison between the current study and previous findings.

For the trials where the treadmill is on, it is unclear whether the animal had to run for 10 or 30 seconds continuously or they were allowed to stop intermittently. Based on the theta bout durations shown in Figure 1C, it seems that the animals can stop during the run. If so, do the animals need to accumulate 10 or 30 seconds of running time, or does the time counter continue even when the animal is not running?

In the Pastalkova et.al., 2008 paper, animals are required to run continuously and steadily over the 10-20 second delay. This difference may explain the difference in the time cell abundance after 5-second delay.

We did not state with sufficient clarity that the running in our task had to be continuous. A difference in running pattern can therefore not be an explanation for differences to Pastalkova et al. (2008). The treadmill was already on when the rat entered and still on when the rat exited. For the rat to hold a steady position (as now shown in Fig. 1b and c, Fig. S1), there was therefore no other option than running continuously at the treadmill speed. Is there an explanation for theta bouts during treadmill-on trials that are shorter than the full extent of the delay? We used a conservative definition of bout length, and examples are now included that show that presumptive pauses in theta are often caused by just one or few cycles that don't show stereotypical waveforms (Fig. S2).

To address these concerns, we now (1) updated the text to unambiguously state that the running was continuous throughout the entire treadmill-on delay period, (2) included panels that show the running speed and head position throughout the delay interval (Fig. 1b and c, Fig. S1), (3) quantified the total time with theta oscillations, which approaches the entire delay length in treadmill-on trials (Fig. 1i), (4) quantified the number of trials where our conservative analysis shows continuous theta during most of the delay period (Fig. 1j), (5) added numerous examples that compare raw traces and our quantification (Fig. S2), and (6) analyzed time cells in only trials in which our conservative criterion for theta bouts shows continuity throughout the delay periods (Fig. S9).

Line 59-64:

'To test to what extent the neural code in delay intervals during a WM task is determined by ongoing theta oscillations, we designed a delayed alternation task with trials when rats were either forced to run continuously on a treadmill ('treadmill on') or were allowed to rest during the delay interval ('treadmill off'; Figure 1a). When running continuously, the head remained in a consistent position, and by design, running speed throughout the entire delay interval corresponded to the treadmill speed (Figure 1b and c, Figure S1).'

See above for Fig. S1 and Fig. 1c

Fig. S9:

Another related question: In the treadmill-on condition, the theta amplitude drops after ~5 seconds. How much of this drop is related to running speed change over the delay period? What is the running speed profile over the delay period? In the treadmill-off block, the theta amplitude drops after 2-3 seconds on average. Can the firing fields preferentially appear in trials where the initial theta bout is longer?

The high theta amplitude at the beginning of the treadmill-on trial is a consequence of a jump/fast run at entry into the delay zone—the treadmill was already on and to make forward progress towards the barrier, the rats had to exceed the treadmill speed. Once the head was in proximity to the barrier, the rats were running at the treadmill speed for the remainder of the trial. In treadmill-off trials, the rats also ran into the delay zone before settling down for the remainder of the delay. Higher theta amplitudes are therefore in both cases related to more vigorous locomotion at the entry into the delay. This resulted in a high theta amplitude, which was maintained for the remainder of the delay with the treadmill on and which dropped substantially after the first 1-2 seconds with the treadmill off (Fig. 1e and f).

Fig. 1e and f:

Line 74-82:

'High-amplitude theta oscillations were observed in all 5 rats during forced running throughout the 10-s and 30-s delay intervals (Figure 1e-h, Figure S2 and S3), such that theta power was higher and theta bouts were longer when rats were continuously running with the treadmill on compared to resting with the treadmill off (z-scored theta power, on/10-s: 0.53 ± 0.06 , off/10-s: -0.05 ± 0.07 ; on/30-s: 0.55 ± 0.07 , off/30-s: -0.01 ± 0.06 , mean \pm SEM, $n = 18$ sessions in 5 rats; treadmill on vs. off, 10-s delay: $p = 1.1 \times 10^{-7}$; 30-s delay: $p = 7.7 \times 10^{-12}$, paired t -tests; all theta bouts, on/10-s: 5.15 ± 0.47 s, off/10-s: 2.79 ± 0.19 s, on/30-s: 8.54 ± 1.30 s, off/30-s: 3.59 ± 0.51 s, mean \pm SEM, $n = 18$ sessions; $F(3,68) = 12.51$, $p = 1.3 \times 10^{-6}$, ANOVA; initial theta bouts, on/10-s: 5.28 ± 0.45 s, off/10-s: 2.86 ± 0.28 s, on/30-s: 7.94 ± 1.18 s, off/30-s: 3.37 ± 0.51 s, mean \pm SEM, $n = 18$ sessions; $F(3,68) = 11.66$, $p = 3 \times 10^{-6}$, ANOVA).'

To now better show the profile of the theta amplitude and the running speed throughout the delay, we added panels to the main figure (Fig. 1b, e, f) and added supplementary figures (Fig. S1 and 2) that show these measurements during treadmill-on and treadmill-off trials. We show the average and also a few examples of

individual trials. To answer the question whether some firing fields preferentially appear in trials with longer theta bouts, we now selected trials that show continuous theta, and separately plot and analyze these data. Even in these highly selected trials, we do not observe evidence for well-timed neuronal firing beyond the first few seconds (Fig. S9). These results are now added to the text. See above for figures.

Line 228-231:

'In addition, we checked whether it might require long periods of uninterrupted theta in treadmill-on conditions to elicit more prolonged time cell sequences and examined only trials with theta bouts over at least 80% of the delay period. Again, the same pattern as for the complete dataset emerged, with peak times of time-limited cells predominantly in the first few seconds (Figure S9).'

In the MacDonald et. al., 2014 time cell paper, time cells are present beyond the 5-second delay, even though the animal is not moving much in the delay area. In that study, theta oscillations existed throughout the delay period. This finding, together with the results from the Pastalkova 2008 paper, implies that potentially continuous theta might play a role in producing continuous sequences across the delay period.

Based on these papers, which we candidly describe in our manuscript, we hypothesized that we would find the same. However, despite including recordings from a larger number of rats in our study (n = 13 in the original manuscript with n = 6 added in the revised version; total n = 19) compared to previous publications (n = 3 in Pastalkova et al., n = 3 in Wang et al., and n = 4 in MacDonald et al.) we do not find this effect in any of our rats. As mentioned above, a reduced continuity in running/theta is not a viable explanation for the difference. Of these papers, the one with the most similar behavioral task to our study is the task in Pastalkova et al. (2008), and close inspection of their data shows that there was also an overrepresentation of time cells in the first few seconds of the delay interval (their Fig. 5).

In addition, both papers mentioned above show clear differential firing between either left- and right-turn trials or trials with different object and odor pairs. In the Pastalkova et.al. paper, the animals were first trained for 3-second wheel running before starting the alternation task training. In this manuscript and the Ito et.al., 2015 paper, it seems that the animals were trained for alternation before introducing the delay. It is unclear whether the difference in training protocols might lead to differences in the occurrence of turn-selective firing cells.

We agree that differences in training protocols may result in the differences in the likelihood that differentially active cells are found. However, we would like to point out that the finding of differentially/sequentially active cells was not even equivocal in the original Pastalkova et al. study. Their figures show that the effect varied across the three rats in the study, with most differentially active cells early in the delay interval. It would appear that Wang et al. reproduced the findings of Pastalkova et al. in an additional 3 rats, but close inspection of the behavioral procedures across studies shows that Wang et al. required the rats to turn around at the goal location and to return to the running wheel along the same path. Although this is a small modification, it makes the task more akin to a linear track with a running wheel at its center location, which may make it more likely that the wheel is taken as part of a continuous path, where it is common that hippocampal cells are differentially active across running directions (e.g., McNaughton et al., 1983; Muller et al., 1994).

Furthermore, the seminal paper on splitter cells by Wood et al. (2000) showed that differentially active cells are particularly abundant in a continuous alternation task, which would be an argument against the assumption that this type of training would preclude the emergence of turn-selective cells in the stem/delay. As mentioned, Ito et al. (2015) also did not find turn-selective cells in the hippocampus in a delayed spatial alternation task. Together with showing the effect in a large number of rats in our studies (7 in Sabariego et al., 13 in the original submission, and an additional 6 rats added in the revised version; total n = 26), it therefore seems reasonable to propose that the lack of differential activity in the rodent hippocampus is not an uncommon finding, which the research community should take into account when devising models of working memory.

Irrespective of whether there exists a training protocols in which the likelihood of differentially active cells is higher than in our protocol, our data show that rats can perform delayed spatial alternation with high accuracy (~90% correct) without this type of firing pattern. Therefore, differential delay activity cannot be a universal mechanism for working memory retention. Similar to the series of studies in rodents, seminal work has shown differentially active cells in working memory tasks in non-human primates in prefrontal cortex (e.g., Fuster and Jervey, 1981), but more recent studies have shown that this is not the only, if even the most predominant, mechanism for working memory retention (Stokes, 2015, 19, 394-405; Trends Cogn Sci; Miller et al., 2018, Neuron, 100, 463-475).

To more broadly address the discrepancies across studies in rodents, we have now expanded this section of the discussion.

Line 371-398:

'Relatedly, an additional unexpected finding in our data is that neither the early time cells nor the later persistently active cells were informative about turn direction, which is conflicting with some previous reports^{8,9}, but not the only report of either a low or non-significant fraction of turn-selective cells in delay intervals without running^{13,16,20}. Here, we directly tested whether running and its associated continuity of theta oscillations are a possible source for the discrepancy between studies with and without prolonged sequential firing patterns^{8,9,16} by including blocks of trials with and without running in our experiment. There were no major differences in the fraction of time cells or in the length of the time period with sequential firing patterns across these conditions. This raises the question whether other differences across studies could explain the discrepancy. First, we tested whether the criterion for identifying time cells could influence the result, which was not the case. Second, we tested whether the extent of familiarity leads to more memory-related firing. We found a higher proportions of time cells with less training, but time cells were nonetheless not turn-selective. Third, we included the initial entry path into the delay zone in the analysis and found an increase in the proportion of turn-selective cells. While this is perhaps not surprising, considering that the path had not converged to a common behavioral pattern across right-turn and left-turn trials, it is consistent with the finding that most differentially active cells are found early in the delay period⁸. Finally, our training protocol included a phase with continuous spatial alternation, and such training could perhaps reduce differential firing later in the task. However, turn-selective firing is particularly abundant in the continuous task version²⁷, which makes it unlikely that training in this task variant occludes turn-selective firing. Similarly, a task variant with a high proportion of turn-selective cells requires running back and forth in opposite directions along a common path. This behavioral pattern resembles running back and forth on a linear track, which is known to result in direction-selective firing^{31,32} that may extend to the included running wheel¹⁸. Although there are thus examples of experimental conditions in which two separate hippocampal firing patterns emerge across choices for which WM is required^{8,33-35}, these findings do not generalize. Sequential and memory-related hippocampal firing patterns during delay intervals are not consistently found in hippocampus-dependent WM tasks, including in the large sample in our study (n = 19 rats). In particular, sustained theta oscillations during the delay interval do not necessarily result in memory-related firing patterns during the delay.'

There also seems to be a large difference in terms of the percentage of neurons with fields in the delay area, between the current study (~ 10%) and the Pastalkova (32%), or MacDonald (53%) papers. The percentage of time cells is even lower in the animals trained specifically for the treadmill on task. What might be the reasons for this discrepancy? Do the time cells recorded in the current study show clear phase precession?

It is correct that we see a lower proportion of time cells, which is perhaps not unexpected given that we do not find that the time-cell activity extended beyond 5 s. Furthermore, we now performed analysis of the proportion of time cells by using the classification criterion of Pastalkova et al. (2008). This yields an even lower proportion of time cells (Fig. S8a), and the classification scheme can therefore not be the source of the discrepancy. We did

not perform a direct comparison with MacDonald et al. because of the major differences in task design and because the information in their methods were not sufficiently detailed to reproduce their criterion.

However, we identified that more extensive training is one of the factors that explains the different proportions of time cells. To address this point, we included new data (n = 6 rats) in which the same behavioral design as for the original data was used, but recordings were taken after fewer training days. Less training resulted in a higher proportion of time cells (Fig. S8b). Nonetheless, the result that time cells predominantly covered only the first few seconds of the delay period and were generally not turn-selective held up irrespective of training extent and proportion of time cells.

These new analyses are now presented in the text and in Fig. S8.

Fig. S8:

Line 205-217:

'We next examined whether the low fraction of time-limited cells and the early peak times in our study may have been the consequence of a more rigorous criterion than in earlier studies, and for comparison, applied the criterion from ref. 8 to our data. With their criterion, we detected an even lower fraction of time cells than with our shuffling method (4.9-7.7% across treadmill/delay conditions, n = 5 rats, Figure S8a and see Table S3 for statistics), and again, the cells that were detected included a high fraction of cells with peak times in only the first few seconds (84.9-100.0%). Finally, we reasoned that the substantial differences in the fraction of time cells compared to other studies could be a consequence of more extensive training experience. Therefore, we added an additional experimental group (n = 19 sessions in 6 rats). In these rats, we recorded CA1 cells with chronically implanted Neuropixels probes, which allowed us to perform recordings with a more limited number of training days. These recordings revealed a much higher proportion of time cells in rats with less training (38.1-54.6% across rats, n = 6 rats, compared to 15.0-20.0%, n = 5 rats; Figure S8b and c), and again, included a high fraction of time-limited cells of which most had their peak times in the first few seconds (90.9-100.0%).'

We now measure the phase precession of time cells during the delay in treadmill-on conditions, where theta oscillations are continuous. A substantial proportion of time cells shows theta phase precession. However, the

fraction of persistently active cells with phase precession (~20 %) was substantially lower than fraction of time-limited cells with phase precession (~50 %) (Fig. S10), consistent with the notion that cells that are not sequentially active on the behavioral time scale are also less likely to be sequentially organized within theta cycles.

Fig. S10:

There appear to be more cells persistently active in the 10-second delay trials compared to the 30-second delay trials. If persistently active cells are simply an extension from the time cells, shouldn't we expect to see a similar proportion of cells in the 10-second and 30-second conditions?

Because persistently active cells were previously not described in hippocampus in delayed alternation tasks, we used a conservative criterion to identify them. A cell was classified as persistently active if its firing rate exceeded 20% of its peak rate in every 150 ms bin from the peak time to the end of the delay. Because of the higher number of bins in the 30-s delay condition compared to the 10-s condition, this criterion is even more stringent with longer delays and likely results in lower proportions of positively identified persistently active cells with longer delays.

To more rigorously examine whether cells that were identified in the 10-s condition as persistently active (i.e., firing until the end of the 10-s delay) were also active until the end of the 30-s delay, we generated a new main figure (Fig. 6) that focuses on examples and quantification of persistently active cells. In particular, we quantified the firing rates of persistently active cells across delay conditions. These data show that that the firing rates of persistently active cells at the end of the 10-s delay and at the end of the 30-s delay were strikingly similar ($r = 0.829$, $n = 37$, $p < 0.001$, Spearman correlation). These new results are reported in the text, and as mentioned above, in the new main figure (Fig. 6e).

Fig. 6:

Line 239-243:

'To examine whether cells that were identified in the 10-s condition as persistently active (i.e., firing until the end of the 10-s delay) were also active until the end of the 30-s delay, we quantified the firing rates of persistently active cells across delay conditions. These data show that the cells' firing rates at the end of the 10-s delay and at the end of the 30-s delay were strikingly similar ($r = 0.829$, $n = 37$, $p < 0.001$, Spearman correlation; Figure 6e).'

If I understand this correctly, in Figure 4A, the bottom row contains all the persistently active cells. It looks like these cells can start firing at any time during the delay, not necessarily after 5 seconds. Also, if the median peak time is 4.7 seconds, it is not very convincing to claim that persistently active cells mainly appear after 5 seconds.

In this case, we had reported a median onset time of 4.7 s, not a median peak time of 4.7 s. This was perhaps not obvious, as we reported peak times earlier in the manuscript. To now more explicitly point out that we are reporting onset time, we are also including the criterion in the sentence, which now reads:

Line 243-245:

'Persistently active cells typically turned on after ~5 s (median onset time: 4.7 s, with onset time defined as time when first exceeding 20% of peak rate) and then continued to be active for the remainder of the delay interval.'

From S6, it looks like there are still some theta oscillations in the treadmill off condition.

Yes, there are certainly theta oscillations in the treadmill-off condition, and the key feature of the treadmill-off condition is theta oscillations are often interrupted rather than completely absent. We now added additional analyses and figures to show that effect (Fig. 1e, i and j, Fig. S2), and we rewrote the text to better emphasize that the main difference between the treadmill-on and treadmill-off condition is the continuity, not the presence of theta oscillations.

See above for Fig. 1e, i, and j and for Fig. S2

Line 72-88:

'Because of the link between movement and the emergence of hippocampal theta, we expected that running on the treadmill controlled the power and persistence of theta oscillations during the delay interval. High-amplitude theta oscillations were observed in all 5 rats during forced running throughout the 10-s and 30-s delay intervals (Figure 1e-h, Figure S2 and S3), such that theta power was higher and theta bouts were longer when rats were continuously running with the treadmill on compared to resting with the treadmill off (z-scored theta power, on/10-s: 0.53 ± 0.06 , off/10-s: -0.05 ± 0.07 ; on/30-s: 0.55 ± 0.07 , off/30-s: -0.01 ± 0.06 , mean \pm SEM, $n = 18$ sessions in 5 rats; treadmill on vs. off, 10-s delay: $p = 1.1 \times 10^{-7}$; 30-s delay: $p = 7.7 \times 10^{-12}$, paired t-tests; all theta bouts, on/10-s: 5.15 ± 0.47 s, off/10-s: 2.79 ± 0.19 s, on/30-s: 8.54 ± 1.30 s, off/30-s: 3.59 ± 0.51 s, mean \pm SEM, $n = 18$ sessions; $F(3,68) = 12.51$, $p = 1.3 \times 10^{-6}$, ANOVA; initial theta bouts, on/10-s: 5.28 ± 0.45 s, off/10-s: 2.86 ± 0.28 s, on/30-s: 7.94 ± 1.18 s, off/30-s: 3.37 ± 0.51 s, mean \pm SEM, $n = 18$ sessions; $F(3,68) = 11.66$, $p = 3 \times 10^{-6}$, ANOVA). In addition, delay intervals included longer periods with sustained theta oscillations when the treadmill was on rather than off during the delay (% time with theta: on/10-s: $81.49\% \pm 3.24$, off/10-s: $66.73\% \pm 2.89$, on/30-s: $83.91\% \pm 2.94$, off/30-s: $67.87\% \pm 2.99$, mean \pm SEM, $n = 18$ sessions; $F(3,68) = 9.32$, $p = 3 \times 10^{-5}$, ANOVA; percent trials with > 80% of delay duration in theta: on/10-s: $64.97\% \pm 7.13$, off/10-s: $29.26\% \pm 5.36$, on/30-s: $71.32\% \pm 8.06$, off/30-s: $24.17\% \pm 6.95$, mean \pm SEM, $n = 18$ sessions; $F(3,68) = 12.81$, $p = 1 \times 10^{-6}$, ANOVA; Figure 1i and j).'

Figures 5E and 5F are in agreement with the report in the Ito et.al. 2015 paper on that more turn-selective cells occur in the stem area than in the delay area.

Yes, and this is now more explicitly mentioned in the discussion.

Line 411-416:

'Second, we analyzed cell assemblies and found a high number of distinct assemblies in the delay zone. While assembly activation was not informative when occurring during the delay, we found that their reactivation on the stem—after delay periods without running—was highly predictive of turn direction. Similar findings have previously also been reported for single cells²⁰, but in our single cell analysis, the effect was less pronounced (see Figure 7g). For some types of delay intervals, we therefore found a possible hippocampal contribution to re-instantiating memory-related firing patterns.'

Reviewer #3 (Remarks to the Author):

This manuscript examined the activity of hippocampal cells in delay periods in between spatial working memory trials on a continuous T-maze. Two alternative delay periods of 10 or 30 seconds were used, and animals either ran on a treadmill or remained stationary. They were able to differentiate two different types of cells: those that fired within 5sec with a relatively fixed interval from the delay start and another that exhibited persistent firing during the delay (delay active cells). Cell populations that were active in treadmill running and non-running conditions were different. Surprisingly, the activity of the cells during the delay did not predict the future arm choice of the animal, suggesting that the hippocampus was not involved in ‘rehearsing’ short-term spatial memory in this task during the delay. The work resolved multiple unresolved issues related to ‘time cells’, including the dependence of theta oscillations and delay length. Importantly, it demonstrated that delay time-dependent activity could be observed during relatively shorter delay of <5 s. Overall, I have a few technical questions:

Major comments

Was the lateral position of the animal controlled for when it was examined whether, at the choice point, the activity of cells predicted which arm the animal chose?

In the analysis for predicting turns, we only included the segment of the stem before the path diverged. In the previous analysis, this point was estimated by visual inspection of each recording session. We are now using a quantitative criterion, which is that only segments where the path diverged by <2 cm from the midline were included. With this criterion, we no longer observe a significant proportion of turn-selective cells in the stem segment before overt turning for any of the delay/treadmill conditions.

We assume that the comment also asks whether we could start detecting rate differences between left and right turns once the head position diverged. Therefore, we now also include analysis from the entire length of the stem and are comparing the average path of all left-turn and of all right-turn trials. We show that lateral head position did not systematically differ between the two turn directions until ~50 cm along the stem. There is no turn-selective firing up to this point in the stem. However, differentially active cells begin to emerge once the path starts diverging. We start detecting a significant fraction of left vs right selective cells (55 cm from the delay zone, path diverges -2.5 cm to the left, 2.3 cm to the right, 9.6% of 104 cells are differentially active, $p = 0.036$, binomial test compared to chance level). These results are now described in the text and shown in a new figure panel (Fig. 7g).

Fig. 7g:

Line 310-317:

'Because we could not consistently detect turn-selective cells or cell assemblies during the delay period, we tested whether cells or cell assemblies may be selectively reactivated during the brief period on the stem, after the exit from the delay zone. We restricted the analysis to a segment on the center arm before the path diverged by more than 2 cm from the midline. With this criterion, we did not observe a significant proportion of turn-selective cells for any of the delay/treadmill conditions. Once the path starts diverging (55 cm from the delay zone, -2.5 cm to the left, 2.3 cm to the right), a significant fraction of cells was left vs right selective (Figure 7g; 9.6% of 104 cells, $p = 0.036$, binomial test compared to chance level).'

I assume interneurons were not included in the delay-active cells. At least in Fig2e, some of these had ~10Hz rates. Also, it would be useful to show that the firing rate of these cells drops during maze running.

Yes, it is correct that interneurons were not included. We included cells with an average rate between 0.1 Hz and 5 Hz in the entire recording session, which is mentioned in the methods section.

Line 540-541:

'Clusters were classified as principal cells if their average firing rates were between 0.1 and 5 Hz.'

In addition, new plots are included that show the activity of delay-active cells throughout the maze (Fig. 2c). These plots show that many delay-active cells have a place field outside of the delay zones, but are otherwise active at low rates. Moreover, the lower activity of the delay-active cells outside of the delay zone is also captured by including the 3 s before entry and the 3 s after exit from the delay zone in the plots that show the firing rate throughout the delay intervals (Fig. 2d). Therefore, the relatively high average rates of a subset of delay-active cells are specific to the delay zone, when these cells were persistently active.

Fig. 2c and d:

Was it quantified whether the rates of the 'time cells' were similar at the 10 vs. 30 s delay conditions? Were the time intervals also similar when they were active during the delay? Perhaps it was quantified, but I cannot find an analysis related to this on the figures.

We calculated correlation coefficients between rate vectors (each entry is the firing rate of a cell within a 5 s bin) between the 10-s and 30-s delay conditions, and high correlations for corresponding time bins were observed (Fig. 4c and d). In addition, we had also shown that the peak times of time limited cells were corresponding (previous Fig. S4d, now Fig. S6d). We now also show that the average firing rates over the delay interval of delay-active cells (i.e., all cells with rates > 0.5 Hz within the delay) and of persistently active cells are matching across the 10-s and 30-s delay conditions (Figs. 3e and 6e). Below (Fig. R1), we show the correlation for all time cells (i.e., time-limited cells and persistently active cells) across the 10-s and 30-s conditions, and the results for time cells are similar as for all delay-active cells. To not be redundant, we only include the plots for delay-active cells in the manuscript (Fig. 3e), and the text now refers to these correlations.

Fig. 4c and d:

Fig. S6d:

Fig. 6e:

Fig. 3e:

Fig. R1:

Line 126-130:

'Similar to the proportions, the firing rates of active cells were more similar across delay conditions than across treadmill conditions (difference over sum, 0.22 ± 0.01 and 0.5 ± 0.02 , $p < 0.001$, Kolmogorov-Smirnov test; mean firing rates, 10-s vs. 30-s delay, treadmill on: Spearman's $r = 0.91$, treadmill off: $r = 0.84$; treadmill on vs. off, 10-s delay: $r = 0.18$; 30-s delay: $r = 0.10$; correlation between delay conditions vs. between treadmill conditions, $p < 0.001$, Fisher z-transformation; Figure 3d and e).'

Line 151-155:

'Cells with narrow time fields early in the 10-s delay interval showed approximately corresponding activity patterns early in the 30-s delay interval. As a consequence, the firing rates during the first 5 s of the 10-s delay and the 30-s delay were highly correlated (treadmill on: $r^2 = 0.61$, Pearson's linear correlation; treadmill off: $r^2 = 0.68$, Pearson's linear correlations; Figures 4c and d).'

Line 239-243:

'To examine whether cells that were identified in the 10-s condition as persistently active (i.e., firing until the end of the 10-s delay) were also active until the end of the 30-s delay, we quantified the firing rates of persistently active cells across delay conditions. These data show that the cells' firing rates at the end of the 10-s delay and at the end of the 30-s delay were strikingly similar ($r = 0.829$, $n = 37$, $p < 0.001$, Spearman correlation; Figure 6e).'

Would it be possible to check whether activity in the delay periods during error trials was similar to that of correct trials? This could provide additional evidence that hippocampal activity during the memory delay is not involved.

We now performed analysis of similarity in average firing rates during the delay between correct and error trials and compared it to the baseline similarity between pairs of correct trials. The similarity score across trials was calculated by taking the absolute value of the difference over the sum of the firing rates. The analysis revealed that firing rate differences to error trials were higher than the baseline in 10-s delay, but not in 30-s delays (Fig. S11a and b). This result was consistent across the original dataset (n = 5 rats) and the added dataset (n = 6 rats). This result is now discussed by speculating that the consistency in neuronal firing patterns may be relevant across short delay durations even though neuronal firing patterns are not informative.

Figure S11a and b:

Line 282-286:

'Finally, we performed analysis of similarity in average firing rates during the delay between correct and error trials and compared it to the baseline similarity between pairs of correct trials. The analysis revealed that firing rate differences to error trials were higher than the baseline in 10-s delays, but not in 30-s delays. This result was consistent across two datasets (n = 5 and 6 rats; Figure S11a and b).'

Line 424-428:

'While we thus found only limited information about turn-direction, we identified higher trial-by-trial inconsistency in hippocampal firing rates during 10-s delay intervals for error compared to correct trials. While the hippocampal activity during the delay may therefore not be informative about turn-direction, consistent firing patterns across trials (see Figures S11a and b) are—for short delay durations—indicative of upcoming correct choices.'

This may have been mentioned in the results, but did this study observe spitter cells in the central arm? Some work (e.g., Emma Wood's) did not observe spitter cells during delayed alternation tasks, while others have seen this. The late Howard Eichebaum suggested that depending on how animals are trained, animals may follow predetermined paths from the delayed periods on, or they may decide only at the decision point which way they go next. The strategy of the animal could be inferred based on the presence of spitter cells or testing whether the animals consistently stopped at the choice point.

Similar to what is shown in the seminal Wood et al. paper, we now include analysis of turn-selective firing patterns along spatial bins on the stem. This analysis did not reveal turn-selective firing patterns up to the points where the paths for left-turn and right-turn trials diverge (Fig. 7g). In addition, we analyzed the running speed of our rats (Fig. R2) and did not see any evidence that they stopped at the choice point. To the contrary, they quickly accelerated out of the delay zone and maintained a running speed of ~50 cm/s at the choice point, which transiently increased when the rat's head moved through the turn. It is clear that animals do not stop at the choice point to decide which direction to take. The data therefore do not seem to support the suggestion that rats decide or correct their behavior at the choice point, as previously suggested for animals that exhibit vicarious trial and error (Redish, 2016, Nat Rev Neurosci 17, 147-59). This is consistent with the notion that well-trained rats make the choice before the T-junctions in the maze.

Fig. R2:

Minor

Although the results are clearly written, it is quite boring to read, partially because many of the less important results are discussed in the main text without figures. Perhaps making supplementary figures about less relevant results and only briefly referring to them in the main text would make the text more concise and convey the underlying story better.

We carefully read through the text and moved points that are less essential to supplementary figures, along with shortening the detail provided in the main text. In addition, we removed the previous main Fig. 8 and all corresponding text, which was not sufficiently conclusive, as pointed out by reviewer #4. This was replaced by more detailed analysis of SWRs/population events, as requested by reviewers #1 and #4. The shortening of the description of less relevant results allowed us to include important responses to reviewer comments in the main text without substantially lengthening the main text.

Reviewer #4 (Remarks to the Author):

In this study, Yuan and colleagues set out to study hippocampal activity during the delay interval of a working memory task and its dependence on theta oscillations. They use a behavioral design which is very well suited to address their question: a spatial alternation task with interleaved blocks of “treadmill on” and “treadmill off” conditions, as well as 10- vs 30-second delay durations. They show that sequential firing of neurons in CA1 during a delay period before choice is not influenced by the amount and power of theta oscillations. They also convincingly show that sequential firing takes place in the first few seconds of the delay, while cells active afterwards tend to remain active for the rest of the delay period. They then provide evidence suggesting the absence of trajectory-dependent coding during the delay interval both when mice are required to run and when they are not. Previous work from the same group (Sabariego, Marta et al. 2019) had previously showed that time cells active during the 10-s delay period of the same alternation task did not distinguish between left-turn and right-turn trials; the authors replicate this finding in the manuscript for both type of trial conditions and for assembly activity in addition to single cells activity, making this a strong observation. Finally, the authors analyze large population events happening during the delay interval; they propose that activity during these events did not code for turn direction, and is therefore uninformative about past behavior or future choice. However, the evidence shown in support of this conclusion appears for now a bit preliminary; their data shows that sequences detected during the delay are more similar among each other when compared to sequences detected during reward population events. More evidence is needed to justify this conclusion. Overall, this work will be of general interest to the field of. Since the work challenges in part what was found in previous literature, it would be useful for readers to have a more exhaustive discussion of possible reasons for the differences and discrepancies with previous findings.

Specific comments:

-In figure 1B, what is the reason for the use of theta-delta ratio when quantifying the difference in theta power between conditions? Could the authors include theta spectrograms?

The theta-delta ratio was shown to illustrate the differences, but we agree it is dependent on delta and perhaps not the best measurement of changes over time in the theta band. We now more directly use normalized theta power to compare treadmill-on and off trials (Fig. 1f). This also aligns better with the detection of theta bouts, which used mean theta power plus 0.5 times the standard deviation as the threshold to define theta bouts. To better show the difference in the length of theta bouts between treadmill-on and off trials, we now added additional LFP traces (Fig. 1e and Fig. S2). We also added example spectrograms to show the substantial differences in the consistency of theta oscillations between treadmill-on and off trials (Fig. S3).

Fig. 1e and f:

Fig. S2:

Figure S2

Fig. S3:

- In figure 1D and 3D: why is the firing distribution across the maze (and relative quantifications) shown for all animals together? It is more appropriate to show them separately per animal.

We now added separate plots for each animal (Fig. S4c and S5) that are corresponding to the summary panels (originally Fig. 1D, now Fig. 2c; originally Fig. 3D, now Fig. 4c).

Fig. S5:

Fig. S4c:

-Could the authors speculate on how to explain the difference with previous studies? One point to be addressed is whether animals might be overtrained (3-5 weeks of training with continuous alternation + 10 days with delay), and if that could influence the results.

To not only speculate, but also provide additional data, we added a dataset in which rats ($n = 6$) were trained to a lesser extent. The new data show that the proportion of time cells is higher with fewer training sessions (Fig. S8c). Despite the effects of training extent on the fraction of time cells, we nonetheless continue to observe that most time cells have peaks early in the delay period (Fig. S8b) and that they were not turn-selective (Fig. S8d). To report these data and compare them to our observation of the low proportions of time cells in well-trained rats, we now added these data to the results.

Fig. S8b, c and d:

Line 210-217:

‘Finally, we reasoned that the substantial differences in the fraction of time cells compared to other studies could be a consequence of more extensive training experience. Therefore, we added an additional experimental group ($n = 19$ sessions in 6 rats). In these rats, we recorded CA1 cells with chronically implanted Neuropixels probes, which allowed us to perform recordings with a more limited number of training days. These recordings revealed a much higher proportion of time cells in rats with less training (38.1-54.6%, $n = 6$ rats, compared to 15.0-20.0%, $n = 5$ rats; Figure S8b and c), and again, included a high fraction of time-limited cells of which most had their peak times in the first few seconds (90.9-100.0%).’

We also substantially expanded a paragraph in the discussion to compare our findings to those of previous studies.

Line 371-398:

‘Relatedly, an additional unexpected finding in our data is that neither the early time cells nor the later persistently active cells were informative about turn direction, which is conflicting with some previous reports^{8,9},

but not the only report of either a low or non-significant fraction of turn-selective cells in delay intervals without running^{13, 16, 20}. Here, we directly tested whether running and its associated continuity of theta oscillations are a possible source for the discrepancy between studies with and without prolonged sequential firing patterns^{8, 9, 16} by including blocks of trials with and without running in our experiment. There were no major differences in the fraction of time cells or in the length of the time period with sequential firing patterns across these conditions. This raises the question whether other differences across studies could explain the discrepancy. First, we tested whether the criterion for identifying time cells could influence the result, which was not the case. Second, we tested whether the extent of familiarity leads to more memory-related firing. We found a higher proportions of time cells with less training, but time cells were nonetheless not turn-selective. Third, we included the initial entry path into the delay zone in the analysis and found an increase in the proportion of turn-selective cells. While this is perhaps not surprising, considering that the path had not converged to a common behavioral pattern across right-turn and left-turn trials, it is consistent with the finding that most differentially active cells are found early in the delay period⁸. Finally, our training protocol included a phase with continuous spatial alternation, and such training could perhaps reduce differential firing later in the task. However, turn-selective firing is particularly abundant in the continuous task version²⁷, which makes it unlikely that training in this task variant occludes turn-selective firing. Similarly, a task variant with a high proportion of turn-selective cells requires running back and forth in opposite directions along a common path. This behavioral pattern resembles running back and forth on a linear track, which is known to result in direction-selective firing^{31, 32} that may extend to the included running wheel¹⁸. Although there are thus examples of experimental conditions in which two separate hippocampal firing patterns emerge across conditions for which WM is required^{8, 33-35}, these findings do not generalize. Sequential and memory-related hippocampal firing patterns during delay intervals are not consistently found in hippocampus-dependent WM tasks, including in the large sample in our study ($n = 19$ rats). In particular, sustained theta oscillations during the delay interval do not necessarily result in memory-related firing patterns during the delay.'

-Given that the authors themselves speculate, earlier in the manuscript, that interleaving trials with and without running could make a difference, compared to just training animals with one trial type, the analysis shown in fig.5 should be repeated for animal groups with running trials only, in order to make the claim more solid.

We repeated the analysis for groups with only treadmill-on sessions and for the group with only treadmill-off groups and these results are now reported in Table 4. In the treadmill-on only group, during the 10-s delay interval, assemblies carried information about turn direction in the 10-s delay, but not in the 30-s delay. This suggests that running during the delay can render cell assemblies informative, but only over short durations, and this is now reported.

Line 303-308:

'Despite the large pool of assemblies that was detected during each type of delay, assembly activation during the delay was not consistently informative for one or the other turn direction beyond chance levels (Figure 7a and Table S4). We repeated the analysis for groups with only treadmill-on or only treadmill-off training. Only with treadmill-on training, assemblies carried information about turn direction in the 10-s delay, but not in the 30-s delay (Table S4). This suggests that running during the delay can render cell assemblies informative, but only over short durations.'

Fig. 7a:

Table S4. Assembly numbers and turn-selective assemblies in the delay and stem area

		Delay				Stem			
		# of assemblies				# of assemblies			
		active ¹	turn-selective ²	% selective	P ³	active ¹	turn-selective ²	% selective	P ³
on only	on/10-s	30	8	26.7	0.0001	11	2	18.2	0.10
	on/30-s	33	3	9.1	0.23	13	0	0.00	0.51
off only	off/10-s	40	4	10.0	0.14	12	0	0.00	0.54
	off/30-s	50	5	10.0	0.10	11	0	0.00	0.57
on and off	on/10-s	61	4	6.6	0.36	24	2	8.3	0.34
	on/30-s	64	4	6.3	0.40	24	3	12.5	0.12
	off/10-s	95	8	8.4	0.10	27	6	22.2	0.0019
	off/30-s	105	9	8.6	0.080	30	10	33.3	0

¹ active in at least 4 of 20 trials, assemblies were initially detected during the delay but could also activate (strength >1 standard deviation above mean) in other maze segments, ² L/R assembly strength difference > 95% of shuffle, ³ Binomial test.

- In order for results in fig.6 to support the author’s claim that “Large-scale population events in the delay zone did not code for turn direction”, only comparing delay and reward events as a whole does not seem sufficient. For instance, the same analysis of temporal activation patterns could be performed, but using activity in half of the trials before turns in one direction to create their template, and use that as a comparison with the other half, as well as to turns in the other direction.

We agree that this analysis was not exhaustive and now replaced it with analysis that examined whether activity on the right or left return arm was preferentially replayed during population events (as also suggested by reviewer #2). Although we observed a substantial number of place cells on either the left or right arm, we did not observe that these were then again selectively active in population events during the delay in either left-turn or right-turn trials (Fig. 7f). In contrast, population events in the reward area, which are often associated with SWRs, had a high proportion of cells that were selectively reactivated during population events at only one or the other reward location (Fig. 7f).

Fig. 7f:

This result was also observed more broadly when analyzing whether assemblies were differentially active during the delay interval (Fig. 7a), but note that cells could be part of an assembly irrespective of whether they had spatially selective activity elsewhere on the maze.

See above for Fig. 7a

Line 278-282:

'However, we did not find turn-selective reactivation above chance for population events in the delay zone (off/10-s: $p = 0.090$, $n = 92$; off/30-s: $p = 0.064$, $n = 194$). As a control, we analyzed left vs. right selective reactivation at the reward locations and detected clear differences (on/10-s: $p < 0.001$, $n = 86$; off/10-s: $p < 0.001$, $n = 139$; on/30-s: $p < 0.001$, $n = 92$; off/30-s: $p < 0.001$, $n = 86$; Figure 7f).'

Minor comments:

-The difference in the percentage of turn-selective cells and assemblies between the two conditions (treadmill on and treadmill off) for activity during the stem (fig 5F) seems puzzling, given that the behavioral state of the animals should be comparable at that point in time How could that be explained?

This is an important observation, and we now discuss this point by suggesting that short-term plasticity or circuit activity during the preceding delay period determines whether cell assemblies are activated over the next few seconds. As pointed out, the oscillation patterns are subsequently indistinguishable while rats are running on the stem. The most parsimonious explanation is therefore that prior circuit activity or plasticity determines subsequent firing patterns.

Line 415-424:

'For some types of delay intervals, we therefore found a possible hippocampal contribution to re-instantiating memory-related firing patterns. However, the finding that such memory-related cell assembly re-activation occurred predominantly after delay periods without persistent theta suggests that assembly organization on the stem depends on the oscillatory dynamics during the preceding delay interval. For example, it can be speculated that short-term plasticity—either locally or in a larger brain circuit—occurs in treadmill-off trials, which later initiates turn-selective activity of hippocampal assemblies. If these mechanisms specifically occur outside of theta oscillations (e.g., SWRs/population events), the same computations would not be engaged in treadmill-on trials, which then require a different set of computations to support memory retention.'

- Regarding the analysis of population events (fig.6), could the authors justify the reasons why population events were defined as they did, with event onset defined as the first spike time of any neuron active during the event itself, rather than by detecting ripples from the LFP?

Moreover, data regarding the number of populations events during reward and delay epochs (both as a whole and per animal) is missing.

We now included a comparison between the occurrence of sharp-wave ripples and population events, and suggest that a criterion that directly uses the neuronal activity level is better suited to analyze neuronal activity patterns. To better justify our reasoning for considering population events SWR-related, we now report the number of SWRs that are accompanied by population events and vice versa. Most population events are accompanied by SWRs, which is now shown in Fig. 7d and described in the text.

Fig. 7d:

Line 276-278:

'We confirmed that most population events in the delay interval and at the reward locations were accompanied by sharp-wave ripples (SWRs; 54.5%/52.6% in delay/reward; Figure 7d and e).'

Line 624-630:

'To further analyze the temporal activation pattern of principal neurons in the delay and reward areas, we selected recording sessions with at least 20 recorded principal cells and identified population events. A population event was defined as a 200 ms time bin when more than 1/5 of all neurons became active, but with neurons only included in the count if they were silent during the preceding 100 ms window (spike count ≤ 1). We reasoned that a criterion that directly selects brief periods of heightened neuronal activity, as otherwise also characteristic for SWRs, is best suited to analyze neuronal activity patterns.'

In addition, the number of population events during reward and delay is included in the text and figure legend for panel 7f.

Line 278-282:

'However, we did not find turn-selective reactivation above chance for population events in the delay zone (off/10-s: $p = 0.090$, $n = 92$; off/30-s: $p = 0.064$, $n = 194$). As a control, we analyzed left vs. right selective reactivation at the reward locations and detected clear differences (on/10-s: $p < 0.001$, $n = 86$; off/10-s: $p < 0.001$, $n = 139$; on/30-s: $p < 0.001$, $n = 92$; off/30-s: $p < 0.001$, $n = 86$; Figure 7f).'

Figure 7 legend:

'f. Population event-associated firing rates were significantly turn-selective in the reward area (left; on/10-s, $p < 0.001$, $n = 86$; off/10-s, $p < 0.001$, $n = 139$; on/30-s, $p < 0.001$, $n = 92$; off/30-s, $p < 0.001$, $n = 86$ population events), but not during delay intervals (right; off/10-s, $p = 0.090$, $n = 92$; off/30-s, $p = 0.064$, $n = 194$ population events; chance = 5%, blue dotted line).'

-It would be useful to allow readers to compare, throughout the manuscript, data coming from individual animals, to judge their consistency and variability.

In addition to adding comprehensive panels for the spatial and temporal firing patterns of each animal (Fig. S4c and S5) in response to one of the major comments, we now also show the running speed and the head position of each animal in the delay zone (Fig. S1) and examples of oscillations in the theta range during treadmill-on and off trials from each animal (Fig. S2).

See above for figures.

Reviewer #5 (Remarks to the Author):

I co-reviewed this manuscript with one of the reviewers who provided the listed reports. This is part of the Nature Communications initiative to facilitate training in peer review and to provide appropriate recognition for Early Career Researchers who co-review manuscripts."

Thank you for participating in this opportunity and providing valuable additional insight.

POINT-BY-POINT RESPONSES TO COMMENTS ON THE REVISED VERSION (Yuan et al., NCOMMS-24-30026A)

Point-by-point responses to editor and reviewer comments are included below, with editor and reviewer comments retained in black and our responses added in blue. A revised manuscript with these revisions is included with the resubmission. Please note that we also corrected minor errors, and we list more detail on these corrections after our responses to comments.

We sincerely thank the editor and the five reviewers for their time and expertise dedicated to evaluating our manuscript. We highly appreciate that the reviewers agreed that we clearly addressed their previous comments.

We addressed the remaining comments by reviewer #2 as well as all the points in the editorial checklist. Responses to the editorial checklist are included here as well as in the Author Checklist.

We hope that all remaining points are addressed and that manuscript will now be accepted for publication in *Nature Communications*.

Reviewer comments:

Reviewer #1 (Remarks to the Author):

The authors have clearly addressed this reviewer's questions and provided a substantial amount of data, including additional experiments, to support the queries. The manuscript makes an additional contribution by dissociating the role of time cells in a working memory task.

Reviewer #2 (Remarks to the Author):

The authors have done a great job and significantly improved the manuscript.

One point that may benefit from further discussion is the relationship among running speed, theta oscillations, time cells, and working memory. A substantial body of evidence indicates that both the power and frequency of hippocampal theta oscillations scale positively with running speed. Prior studies that robustly reported time cells, such as Pastalkova et al. (Science, 2008) and MacDonald et al. (Neuron, 2013), involved relatively high running speeds—exceeding 50 cm/s during wheel running and ranging from 35–49 cm/s on a treadmill, respectively. In both cases, time cells were observed across extended delay periods of up to 15 seconds.

Speed could certainly be a factor. It is ambiguous whether the reviewer refers to MacDonald et al. (2011) or to Kraus et al. (2013), both in Neuron. This is of relevance because there is very limited detail on the task in MacDonald et al. (2011), but with a supplementary figure (Fig. S2) showing that time cells are mostly active below 20 cm/s. In addition, their delay was <10 s. In contrast, Kraus et al. (2013) used running speeds up to 49 cm/s and intervals up to 16 s. Therefore, we assume that the Kraus et al. (2013) and Pastalkova et al. (2008) studies are the relevant ones, and these are now mentioned in the discussion.

By contrast, in both the current study and the earlier work by Yong et al. (bioRxiv, 2022), animals maintained a much lower running speed of ~20 cm/s. Notably, these studies concluded that time cells are not essential for successful performance in working memory tasks. The discrepancy in running speed may provide a plausible explanation: slower movement could result in weaker theta engagement, thereby limiting the recruitment or temporal span of time cells, especially beyond 5 seconds. It is therefore plausible that the reduced speed in the present task both lowers cognitive demand and constrains the contribution of time cells to task performance.

We added this point to the discussion, and the relevant section now reads:

Line 361-368:

'Sequential activity patterns thus have a limited lifetime and are not sustained across long delay intervals, even in conditions when theta oscillations are ongoing throughout the delay. However, it is feasible that the high running speeds in some previous studies^{8,11} resulted in more vigorous theta engagement and extended the temporal span of sequential activity to intervals of ~15 s. Irrespective of potential mechanisms that recruit time cells over longer intervals, we show that sequential firing patterns throughout the delay are not necessary for memory retention, including in conditions with uninterrupted theta oscillations. In our data, there were no sustained time cell sequences during the delay interval—with or without sustained theta—when memory was intact.'

Finally, while the observed increase in the proportion of time cells with less training is an interesting result, differences in experience may not fully explain the variability in time cell prevalence. For instance, animals in Pastalkova et al. (2008) also reached stable performance within approximately one week—comparable to the timeline in the current study.

This is exactly the point that we were conveying—fewer time cells with more experience and more time cells with less experience.

To more explicitly make this point, we rewrote the section describing these data:

Previous version:

'Finally, we reasoned that the substantial differences in the fraction of time cells compared to other studies could be a consequence of more extensive training experience. Therefore, we added an additional experimental group (n = 19 sessions in 6 rats). In these rats, we recorded CA1 cells with chronically implanted Neuropixels probes, which allowed us to perform recordings with a more limited number of training days. These recordings revealed a much higher proportion of time cells in rats with less training (38.1-54.6%, n = 6 rats, compared to 15.0-20.0%, n = 5 rats; Figure S8b and c), and again, included a high fraction of time-limited cells of which most had their peak times in the first few seconds (90.9-100.0%).'

New version, line 210-217:

'Finally, we reasoned that the low fraction of time cells compared to other studies could be a consequence of extensive training experience in our cohort. Therefore, we added an additional experimental group (n = 19 sessions in 6 rats). In these rats, we recorded CA1 cells with chronically implanted Neuropixels probes, which allowed us to perform recordings with a more limited number of training days. These recordings revealed a much higher proportion of time cells in rats with less training (38.1-54.6%, n = 6 rats, compared to 15.0-20.0%, n = 5 rats; Figure S8b and c), and again, included a high fraction of time-limited cells of which most had their peak times in the first few seconds (90.9-100.0%).'

Reviewer #3 (Remarks to the Author):

The revision has addressed the issues I have raised and my questions.

Reviewer #4 (Remarks to the Author):

The authors have answered all my comments. This work will have an important contribution to the field.

Reviewer #5 (Remarks to the Author):
